# Weakly Coupled Deep Q-Networks

**Ibrahim El Shar**
Hitachi America Ltd., University of Pittsburgh
Sunnyvale, CA
ibrahim.elshar@hal.hitachi.com

**Daniel Jiang**
Meta, University of Pittsburgh
New York, NY
drjiang@meta.com

## Abstract

We propose *weakly coupled deep Q-networks* (WCDQN), a novel deep reinforcement learning algorithm that enhances performance in a class of structured problems called *weakly coupled Markov decision processes* (WCMDP). WCMDPs consist of multiple independent subproblems connected by an action space constraint, which is a structural property that frequently emerges in practice. Despite this appealing structure, WCMDPs quickly become intractable as the number of subproblems grows. WCDQN employs a single network to train multiple DQN "subagents," one for each subproblem, and then combine their solutions to establish an upper bound on the optimal action value. This guides the main DQN agent towards optimality. We show that the tabular version, weakly coupled Q-learning (WCQL), converges almost surely to the optimal action value. Numerical experiments show faster convergence compared to DQN and related techniques in settings with as many as 10 subproblems, $3^{10}$ total actions, and a continuous state space.

## 1 Introduction

Despite achieving many noteworthy and highly visible successes, it remains widely acknowledged that practical implementation of reinforcement learning (RL) is, in general, challenging [15]. This is particularly true in real-world settings where, unlike in simulated settings, interactions with the environment are costly to obtain. One promising path toward more sample-efficient learning in real-world situations is to incorporate known structural properties of the underlying Markov decision process (MDP) into the learning algorithm. As elegantly articulated by [44], structural properties can be considered a type of "side information" that can be exploited by the RL agent for its benefit. Instantiations of this concept are plentiful and diverse: examples include factored decompositions [33, 10, 47], latent or contextual MDPs [21, 39, 52], block MDPs [14], linear MDPs [32], shape-constrained value and/or policy functions [49, 37, 31], MDPs adhering to closure under policy improvement [8], and multi-timescale or hierarchical MDPs [23, 13], to name just a few.

In this paper, we focus on a class of problems called *weakly coupled MDPs* (WCMDPs) and show how one can leverage their inherent structure through a tailored RL approach. WCMDPs, often studied in the field of operations research, consist of multiple subproblems that are independent from each other except for a coupling constraint on the action space [24]. This type of weakly coupled structure frequently emerges in practice, spanning domains like supply chain management [24], recommender systems [65], online advertising [9], revenue management [53], and stochastic job scheduling [63]. Such MDPs can quickly become intractable when RL algorithms are applied naively, given that their state and action spaces grow exponentially with the number of subproblems [45].

One can compute an upper bound on the optimal value of a WCMDP by performing a Lagrangian relaxation on the action space coupling constraints. Importantly, the weakly coupled structure allows the relaxed problem to be *completely decomposed* across the subproblems, which are significantly easier to solve than the full MDP [24, 1]. Our goal in this paper is to devise a method that can integrate the Lagrangian relaxation upper bounds into the widely adopted value-based RL approaches of Q-learning

37th Conference on Neural Information Processing Systems (NeurIPS 2023).

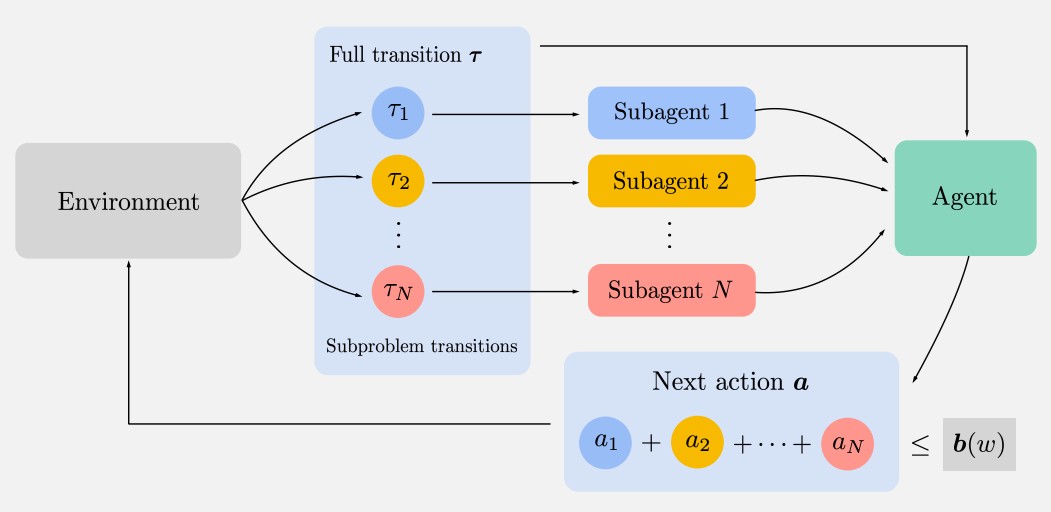

Figure 1: An illustration of our RL approach for WCMDPs. Our approach takes a single "full" transition $\tau$ (as collected by a standard RL agent) and decomposes it into subproblem transitions $\tau_i$ that are passed to "subagents," which are powered by a single network and aim to solve the easier subproblems. The results of these subagents are then used collectively to guide the main agent toward the optimal policy, whose actions need to satisfy *linking constraints*. Here, we illustrate the case of a single linking constraint that requires the sum of the actions to be bounded by a right-hand-side quantity $\boldsymbol{b}(w)$, where $w$ is an exogenous state from the environment.

[59] and deep Q-networks (DQN) [43]. Our proposed method is, to our knowledge, the first to explore the use of Lagrangian relaxations to tackle general WCMDPs in a fully model-free, deep RL setting.

**Main contributions.** We make the following methodological and empirical contributions.

1. First, we propose a novel deep RL algorithm, called *weakly coupled deep Q-networks* (WCDQN), that exploits weakly coupled structure by using a set of *subagents*, each attached to one of the subproblems, whose solutions are combined to help improve the performance of main DQN agent; see Figure 1 for a high-level overview.

2. Second, we also propose and analyze a tabular version of our algorithm called *weakly coupled Q-learning* (WCQL), which serves to conceptually motivate WCDQN. We show that WCQL converges almost surely to the optimal action-value.

3. Finally, we conduct numerical experiments on a suite of realistic problems, including electric vehicle charging station optimization, multi-product inventory control, and online stochastic ad matching. The results show that our proposed algorithm outperform baselines by a relatively large margin in settings with as many as 10 subproblems, $3^{10}$ total actions, and a continuous state space.

## 2 Related Literature

**Weakly Coupled MDPs.** This line of work began with [61] under the name of *restless multi-armed bandits* (RMAB), where there are two actions ("active" or "passive") for each subproblem (also known as "project" or "arm"), under a budget constraint on the number of active arms at any given time.[1] As we will see soon, this is a special case of a WCMDP with two actions per subproblem and a single budget constraint. A popular solution approach to RMABs is the *Whittle index* policy, which was first proposed by [61] and uses the idea of ranking arms by their "marginal productivity." The policy has been extensively studied in the literature from both applied and theoretical perspectives [20, 41, 29, 28, 42, 64]. Whittle conjectured in [61] that the Whittle index policy is asymptotically optimal under a condition called *indexability*; later, [60] established that asymptotic optimality requires indexability, but also another technical condition, both of which are difficult to verify.

---

[1]We note that WCMDPs should not be confused with *constrained MDPs*, where a budget constraint is imposed on the overall cost of the policy in all periods [2].

As discussed in detail by [64], relying on the Whittle index policy in real-world problems can be problematic due to hard-to-verify technical conditions (and if not met, computational robustness and the heuristic's original intuitive motivation may be lost).

A number of recent papers have considered using RL in the setting of RMABs, but nearly all of them are based on Whittle indices [19, 46, 34, 35, 3, 50, 62], and are thus most useful when the indexability condition can be verified. Exceptions are [34] and [35], which propose to run RL directly on the Lagrangian relaxation of the true problem to obtain a "Lagrange policy." Our paper is therefore closest in spirit to these two works, but our methods target the optimal value and policy (with the help of Lagrangian relaxation) rather than pursuing the Lagrange policy as the end goal (which does not have optimality guarantees in general). Moreover, compared to the other RL approaches mentioned above, we do not require the indexability condition and our method works for any WCMDP.

Relaxations of WCMDPs can be performed in several different ways, including approximate linear programming (ALP) [1, 11], network relaxation [45], and Lagrangian relaxation [61, 53, 24, 7, 1, 54, 11]. Notably, [1] provided key results for the ALP and Lagrangian relaxation approaches, and [11] gave theoretical justification for the closeness of the bounds obtained by the approximate linear programming and Lagrangian relaxation approaches, an empirical observation made in [1]. Our work focuses specifically on the Lagrangian relaxation approach, which relaxes the linking constraints on the action space by introducing a penalty in the objective.

**DQN and Q-learning.** The Q-learning algorithm [59] is perhaps the most popular value-based tabular RL algorithm [30, 55, 6], and the DQN approach of [43] extends the fundamental ideas behind Q-learning to the case where Q-functions are approximated using deep neural networks, famously demonstrated on a set of Atari games. Unfortunately, practical implementation of Q-learning, DQN, and their extensions on real-world problems can be difficult due to the large number of samples required for learning [44].

Various papers have attempted to extend and enhance the DQN algorithm. For example, to overcome the over-estimation problem and improve stability, [57] proposes *double DQN*, which adapts the tabular approach of *double Q-learning* from [22] to the deep RL setting. The main idea is to use a different network for the action selection and evaluation steps. [51] modifies the experience replay buffer sampling to prioritize certain tuples, and [58] adds a "dueling architecture" to double DQN that combines two components, an estimator for the state value function and an estimator for the state-dependent action advantage function. Other examples include *bootstrapped DQN* [48], *amortized Q-learning* [56], *distributional RL* [5], and *rainbow DQN* [26].

Our approach, WCDQN, is also an enhancement of DQN, but differ from the above works in that our focus is on modifying DQN to exploit the structure of a class of important problems that are otherwise intractable, while the existing papers focus on improvements made to certain components of the DQN algorithm (e.g., network architecture, experience replay buffer, exploration strategy). In particular, it should be possible to integrate the main ideas of WCDQN into variations of DQN without much additional work.

**Use of constraints and projections in RL.** WCDQN relies on constraining the learned Q-function to satisfy a learned upper bound. The work of [25] uses a similar constrained optimization approach to enforce upper and lower bounds on the optimal action value function in DQN. Their bounds are derived by exploiting multistep returns of a general MDP, while ours are due to dynamically-computed Lagrangian relaxations. [25] also does not provide any convergence guarantees for their approach.

In addition, [16] proposed a convergent variant of Q-learning that leverages upper and lower bounds derived using the information relaxation technique of [12] to improve performance of tabular Q-learning. Although our work shares the high-level idea of bounding Q-learning iterates, [16] focused on problems with *partially known transition models* (which are necessary for information relaxation) and the approach did not readily extend to the function approximation setting. Besides focusing on a different set of problems (WCMDPs), our proposed approach is model-free and naturally integrates with DQN.

## 3 Preliminaries

In this section, we give some background on WCMDPs, Q-learning, DQN, and the Lagrangian relaxation approach. All proofs throughout the rest of the paper are given in Appendix A.

## 3.1 Weakly Coupled MDPs

We study an infinite horizon WCMDP with state space $\mathcal{S} = \mathcal{X} \times \mathcal{W}$ and finite action space $\mathcal{A}$, where $\mathcal{X}$ is the *endogenous* part (i.e., affected by the agent's actions) and $\mathcal{W}$ is the *exogenous* part (i.e., unaffected by the agent's actions) of the full state space. We use the general setup of WCMDPs from [11]. A WCMDP can be decomposed into $N$ subproblems. The state space of subproblem $i$ is denoted by $\mathcal{S}_i = \mathcal{X}_i \times \mathcal{W}$ and the action space is denoted by $\mathcal{A}_i$, such that

$$\mathcal{X} = \otimes_{i=1}^N \mathcal{X}_i \quad \text{and} \quad \mathcal{A} = \otimes_{i=1}^N \mathcal{A}_i.$$

In each period, the decision maker observes an exogenously and independently evolving state $w \in \mathcal{W}$, along with the endogenous states $\boldsymbol{x} = (x_1, x_2, \ldots, x_N)$, where $x_i \in \mathcal{X}_i$ is associated with subproblem $i$. Note that $w$ is shared by all of the subproblems, and this is reflected in the notation we use throughout the paper, where $\boldsymbol{s} = (\boldsymbol{x}, w) \in \mathcal{S}$ represents the full state and $s_i = (x_i, w)$ is the state of subproblem $i$. In addition to the exogenous state $w$ being shared across subproblems, there also exist $L$ *linking* or *coupling constraints* that connect the subproblems: they take the form $\sum_{i=1}^N \boldsymbol{d}(s_i, a_i) \leq \boldsymbol{b}(w)$, where $\boldsymbol{d}(s_i, a_i), \boldsymbol{b}(w) \in \mathbb{R}^L$ and $a_i \in \mathcal{A}_i$ is the component of the action associated with subproblem $i$. The set of feasible actions for state $\mathbf{s}$ is given by

$$\mathcal{A}(\boldsymbol{s}) = \left\{ \boldsymbol{a} \in \mathcal{A} : \sum_{i=1}^N \boldsymbol{d}(s_i, a_i) \leq \boldsymbol{b}(w) \right\}. \tag{1}$$

After observing state $\boldsymbol{s} = (\boldsymbol{x}, w)$, the decision maker selects a feasible action $\boldsymbol{a} \in \mathcal{A}(\boldsymbol{s})$.

The transition probabilities for the endogenous component is denoted $p(\boldsymbol{x}' \,|\, \boldsymbol{x}, \boldsymbol{a})$ and we assume that transitions are conditionally independent across subproblems:

$$p(\boldsymbol{x}' \,|\, \boldsymbol{x}, \boldsymbol{a}) = \Pi_{i=1}^N p_i(x_i' \,|\, x_i, a_i),$$

where $p_i(x_i' \,|\, x_i, a_i)$ are the transition probabilities for subproblem $i$. The exogenous state transitions according to $q(w' \,|\, w)$. Next, let $r_i(s_i, a_i)$ be the reward of subproblem $i$ and let $\boldsymbol{r}(\boldsymbol{s}, \boldsymbol{a}) = \{r_i(s_i, a_i)\}_{i=1}^N$. The reward of the overall system is additive: $r(\mathbf{s}, \boldsymbol{a}) = \sum_{i=1}^N r_i(s_i, a_i)$.

Given a discount factor $\gamma \in [0, 1)$ and a feasible policy $\pi : \mathcal{S} \to \mathcal{A}$ that maps each state $\boldsymbol{s}$ to a feasible action $\boldsymbol{a} \in \mathcal{A}(\boldsymbol{s})$, the value (cumulative discounted reward) of following $\pi$ when starting in state $\boldsymbol{s}$ and taking a first action $\boldsymbol{a}$ is given by the action-value function $Q^\pi(\boldsymbol{s}, \boldsymbol{a}) = \mathbb{E}\big[\sum_{t=0}^\infty \gamma^t r(\boldsymbol{s}_t, \boldsymbol{a}_t) \,|\, \pi, \boldsymbol{s}_0 = \boldsymbol{s}, \boldsymbol{a}_0 = \boldsymbol{a}\big]$. Our goal is to find an optimal policy $\pi^*$, i.e., one that maximizes $V^\pi(\boldsymbol{s}) = Q^\pi(\boldsymbol{s}, \pi(\boldsymbol{s}))$. We let $Q^*(\boldsymbol{s}, \boldsymbol{a}) = \max_\pi Q^\pi(\boldsymbol{s}, \boldsymbol{a})$ and $V^*(\boldsymbol{s}) = \max_\pi V^\pi(\boldsymbol{s})$ be the optimal action-value and value functions, respectively. It is well-known that the optimal policy selects actions in accordance to $\pi^*(\boldsymbol{s}) = \arg\max_{\boldsymbol{a}} Q^*(\boldsymbol{s}, \boldsymbol{a})$ and that the Bellman recursion holds:

$$Q^*(\boldsymbol{s}, \boldsymbol{a}) = r(\boldsymbol{s}, \boldsymbol{a}) + \gamma \, \mathbb{E}\Big[\max_{\boldsymbol{a}' \in \mathcal{A}(\boldsymbol{s}')} Q^*(\boldsymbol{s}', \boldsymbol{a}')\Big], \tag{2}$$

where $\boldsymbol{s}' = (\boldsymbol{x}', w')$ is distributed according to $p(\cdot \,|\, \boldsymbol{x}, \boldsymbol{a})$ and $q(\cdot \,|\, w)$.

## 3.2 Q-learning and DQN

The Q-learning algorithm of [59] is a tabular approach that attempts to learn the optimal action-value function $Q^*$ using stochastic approximation on (2). Using a learning rate $\alpha_n$, the update of the approximation $Q_n$ from iteration $n$ to $n + 1$ is:

$$Q_{n+1}(\boldsymbol{s}_n, \boldsymbol{a}_n) = Q_n(\boldsymbol{s}_n, \boldsymbol{a}_n) + \alpha_n(\boldsymbol{s}_n, \boldsymbol{a}_n)\big[y_n - Q_n(\boldsymbol{s}_n, \boldsymbol{a}_n)\big],$$

where $y_n = r_n + \gamma \max_{\boldsymbol{a}'} Q_n(\boldsymbol{s}_{n+1}, \boldsymbol{a}')$ is the *target* value, computed using the observed reward $r_n$ at $(\boldsymbol{s}_n, \boldsymbol{a}_n)$, the transition to $\boldsymbol{s}_{n+1}$, and the current value estimate $Q_n$.

The DQN approach of [43] approximates $Q^*$ via a neural network $Q(\boldsymbol{s}, \boldsymbol{a}; \theta)$ with network weights $\theta$. The loss function used to learn $\theta$ is directly based on minimizing the discrepancy between the two sides of (2):

$$l(\theta) = \mathbb{E}_{\boldsymbol{s}, \boldsymbol{a} \sim \rho}\Big[\big(y - Q(\boldsymbol{s}, \boldsymbol{a}; \theta)\big)^2\Big],$$

where $y = r(\boldsymbol{s}, \boldsymbol{a}) + \gamma \, \mathbb{E}\big[\max_{\boldsymbol{a}'} Q(\boldsymbol{s}', \boldsymbol{a}'; \theta^-)\big]$, $\theta^-$ are frozen network weights from a previous iteration, and $\rho$ is a *behavioral distribution* [43]. In practice, we sample experience tuples $(\boldsymbol{s}_n, \boldsymbol{a}_n, r_n, \boldsymbol{s}_{n+1})$ from a replay buffer and perform a stochastic gradient update:

$$\theta_{n+1} = \theta_n + \alpha_n\big[y_n - Q(\boldsymbol{s}_n, \boldsymbol{a}_n; \theta)\big] \nabla_\theta Q(\boldsymbol{s}_n, \boldsymbol{a}_n; \theta),$$

with $y_n = r_n + \gamma \max_{\boldsymbol{a}'} Q(\boldsymbol{s}_{n+1}, \boldsymbol{a}'; \theta^-)$. Note the resemblance of this update to that of Q-learning.

### 3.3 Lagrangian Relaxation

The Lagrangian relaxation approach decomposes WCMDPs by relaxing the linking constraints to obtain separate, easier-to-solve subproblems [1]. The main idea is to dualize the linking constraints $\sum_{i=1}^{N} \boldsymbol{d}(s_i, a_i) \leq \boldsymbol{b}(w)$ using a penalty vector $\lambda \in \mathbb{R}_+^L$. The result is an augmented objective consisting of the original objective plus additional terms that penalize constraint violations. The Bellman equation of the relaxed MDP in (2) is given by:

$$Q^\lambda(\boldsymbol{s}, \boldsymbol{a}) = r(\boldsymbol{s}, \boldsymbol{a}) + \lambda^\mathsf{T} \left[ \boldsymbol{b}(w) - \sum_{i=1}^{N} \boldsymbol{d}(s_i, a_i) \right] + \gamma \, \mathbb{E}\left[ \max_{\boldsymbol{a}' \in \mathcal{A}} Q^\lambda(\boldsymbol{s}', \boldsymbol{a}') \right]. \tag{3}$$

With the linking constraints removed, this relaxed MDP can be decomposed across subproblems, so we are able to define the following recursion for each subproblem $i$:

$$Q_i^\lambda(s_i, a_i) = r_i(s_i, a_i) - \lambda^\mathsf{T} \boldsymbol{d}(s_i, a_i) + \gamma \, \mathbb{E}\left[ \max_{a_i' \in \mathcal{A}_i} Q_i^\lambda(s_i', a_i') \right]. \tag{4}$$

It is well-known from classical results that any penalty vector $\lambda \geq 0$ produces an MDP whose optimal value function is an upper bound on the $V^*(\boldsymbol{s})$ [24, 1]. The upcoming proposition is a small extension of these results to the case of action-value functions, which is necessarily for Q-learning.

**Proposition 1.** *For any $\lambda \geq 0$ and $\boldsymbol{s} \in \mathcal{S}$, it holds that $Q^*(\boldsymbol{s}, \boldsymbol{a}) \leq Q^\lambda(\boldsymbol{s}, \boldsymbol{a})$ for any $\boldsymbol{a} \in \mathcal{A}(\boldsymbol{s})$. In addition, the Lagrangian action-value function of (3) satisfies*

$$Q^\lambda(\boldsymbol{s}, \boldsymbol{a}) = \lambda^\mathsf{T} \boldsymbol{B}(w) + \sum_{i=1}^{N} Q_i^\lambda(s_i, a_i) \tag{5}$$

*where $Q_i^\lambda(s_i, a_i)$ is as defined in (4) and $\boldsymbol{B}(w)$ satisfies the recursion*

$$\boldsymbol{B}(w) = \boldsymbol{b}(w) + \gamma \, \mathbb{E}\left[ \boldsymbol{B}(w') \right], \tag{6}$$

*with the exogenous next state $w'$ is distributed according to $q(\cdot \,|\, w)$.*

The first part of the proposition is often referred to as *weak duality* and the second part shows how the Lagrangian relaxation can be solved by decomposing it across subproblems, dramatically reducing the computational burden. The tightest upper bound is the solution of the *Lagrangian dual problem*, $Q^{\lambda^*}(\boldsymbol{s}, \boldsymbol{a}) = \min_{\lambda \geq 0} Q^\lambda(\boldsymbol{s}, \boldsymbol{a})$, where $\lambda^*$ is minimizer.

## 4 Weakly Coupled Q-learning

In this section, we introduce the tabular version of our RL algorithm, called *weakly coupled Q-learning* (WCQL), which will illustrate the main concepts of the deep RL version, WCDQN.

### 4.1 Setup

We first state an assumption on when the linking constraint (1), which determines the feasible actions given a state, is observed.

**Assumption 1** (Linking constraint observability; general setting)**.** Suppose that upon landing in a state $\boldsymbol{s} = (\boldsymbol{x}, w)$, the agent observes the possible constraint left-hand-side values $\boldsymbol{d}(s_i, \cdot)$ for every $i$, along with the constraint right-hand-side $\boldsymbol{b}(w) \in \mathbb{R}^L$.

Under Assumption 1, the agent is able to determine the feasible action set $\mathcal{A}(\boldsymbol{s})$ upon landing in state $\boldsymbol{s}$. Accordingly, it can always take a feasible action. In many cases, it is known in advance that the feasible action set is of the *multi-action RMAB* form: there is a single linking constraint (i.e., $L = 1$) and the left-hand-side is the sum of subproblem actions (i.e., $\boldsymbol{d}(s_i, a_i) = a_i$). In that case, Assumption 1 reduces to the following simpler statement, which we state for completeness.

**Assumption 1′** (Linking constraint observability; multi-action RMAB setting)**.** Suppose that we are in a multi-action RMAB setting. When the agent lands in a state $\boldsymbol{s} = (\boldsymbol{x}, w)$, it observes the constraint right-hand-side $\boldsymbol{b}(w) \in \mathbb{R}$.

In the numerical example applications of Section 6, for illustrative simplicity, we choose to focus on single-constraint settings where Assumption $1'$ is applicable. Note that the "difficulty" of WCMDPs is largely determined by the number of subproblems and the size of the feasible set compared to the full action space, not necessarily by the *number* of linking constraints. In each of our example applications, Assumption 1 naturally holds: for example, in the EV charging problem, there are a limited number of available charging stations (which is always observable).

An important part of WCQL is to track an estimate of $Q^{\lambda^*}(\boldsymbol{s}, \boldsymbol{a})$, the result of the Lagrangian dual problem. To approximate this value, we replace the minimization over all $\lambda \geq 0$ by optimization over a finite set of possible multipliers $\Lambda$, which we consider as an input to our algorithm. In practice, we find that it is most straightforward to use $\lambda = \lambda' \boldsymbol{1}$, where $\boldsymbol{1} \in \mathbb{R}^L$ is the all ones vector and $\lambda' \in \mathbb{R}$, but from the algorithm's point of view, any set $\Lambda$ of nonnegative multipliers will do.

We denote an experience tuple for the entire WCMDP by $\boldsymbol{\tau} = (\boldsymbol{s}, \boldsymbol{a}, \boldsymbol{r}, \boldsymbol{b}, \boldsymbol{s}')$. Similarly, we let $\tau_i = (s_i, a_i, r_i, s_i')$ be the experience relevant to subproblem $i$, as described in (4). Note that $\boldsymbol{b}$ is excluded from $\tau_i$ because it does not enter subproblem Bellman recursion.

## 4.2 Algorithm Description

The WCQL algorithm can be decomposed into three main steps.

**Subproblems and subagents.** First, for each subproblem $i \in \{1, 2, \ldots, N\}$ and every $\lambda \in \Lambda$, we attempt to learn an approximation of $Q_i^\lambda(s_i, a_i)$ from (4), which are the $Q$-values of the unconstrained subproblem associated with $\lambda$. We do this by running an instance of Q-learning with learning rate $\beta_n$. Letting $Q_{i,n}^\lambda$ be the estimate at iteration $n$, the update is given by:

$$Q_{i,n+1}^\lambda(s_i, a_i) = Q_{i,n}^\lambda(s_i, a_i) + \beta_n(s_i, a_i)\big[y_{i,n}^\lambda - Q_{i,n}^\lambda(s_i, a_i)\big], \tag{7}$$

where the target value is defined as $y_{i,n}^\lambda = r_i(s_i, a_i) - \lambda^\intercal \boldsymbol{d}(s_i, a_i) + \gamma \max_{a_i'} Q_{i,n}^\lambda(s_i', a_i')$.

Note that although we are running several Q-learning instances, they all make use of a *common experience tuple* $\boldsymbol{\tau}$ split across subproblems, where subproblem $i$ receives the portion $\tau_i$. We remind the reader that each subproblem is dramatically simpler than the full MDP, since it operates on smaller state and action spaces ($\mathcal{S}_i$ and $\mathcal{A}_i$) instead of $\mathcal{S}$ and $\mathcal{A}$.

We refer to the subproblem Q-learning instances as *subagents*. Therefore, each subagent is associated with a subproblem $i$ and a penalty $\lambda \in \Lambda$ and aims to learn $Q_i^\lambda$.

**Learning the Lagrangian bounds.** Next, at the level of the "main" agent, we combine the approximations $Q_{i,n+1}^\lambda$ learned by the subagents to form an estimate of the Lagrangian action-value function $Q^\lambda$, as defined in (5). To do so, we first estimate the quantity $\boldsymbol{B}(w)$ of Proposition 1. This can be done using a stochastic approximation step with a learning rate $\eta_n$, as follows:

$$\boldsymbol{B}_{n+1}(w) = \boldsymbol{B}_n(w) + \eta_n(w)\big[\boldsymbol{b}(w) + \gamma \boldsymbol{B}_n(w') - \boldsymbol{B}_n(w)\big], \tag{8}$$

where we recall that $w$ and $w'$ come from the experience tuple $\boldsymbol{\tau}$, embedded within $\boldsymbol{s}$ and $\boldsymbol{s}'$. Now, using Proposition 1, we approximate $Q^\lambda(\boldsymbol{s}, \boldsymbol{a})$ using

$$Q_{n+1}^\lambda(\boldsymbol{s}, \boldsymbol{a}) = \lambda^\intercal \boldsymbol{B}_{n+1}(w) + \sum_{i=1}^N Q_{i,n+1}^\lambda(s_i, a_i). \tag{9}$$

Finally, we estimate an upper bound on $Q^*$ by taking the minimum over $\Lambda$:

$$Q_{n+1}^{\lambda^*}(\boldsymbol{s}, \boldsymbol{a}) = \min_{\lambda \in \Lambda} Q_{n+1}^\lambda(\boldsymbol{s}, \boldsymbol{a}). \tag{10}$$

**Q-learning guided by Lagrangian bounds.** We would now like to make use of the learned upper bound $Q_{n+1}^{\lambda^*}(\boldsymbol{s}, \boldsymbol{a})$ when performing Q-learning on the full problem. Denote the WCQL estimate of $Q^*$ at iteration $n$ by $Q_n'$. We first make a standard update towards an intermediate value $Q_{n+1}$ using learning rate $\alpha_n$:

$$Q_{n+1}(\boldsymbol{s}, \boldsymbol{a}) = Q_n'(\boldsymbol{s}, \boldsymbol{a}) + \alpha_n(\boldsymbol{s}, \boldsymbol{a})\big[y_n - Q_n'(\boldsymbol{s}, \boldsymbol{a})\big]. \tag{11}$$

where $y_n = r(\boldsymbol{s}, \boldsymbol{a}) + \gamma \max_{\boldsymbol{a}' \in \mathcal{A}(\boldsymbol{s}')} Q_n'(\boldsymbol{s}', \boldsymbol{a}')$. To incorporate the bounds that we previously estimated, we then project $Q_{n+1}(\boldsymbol{s}, \boldsymbol{a})$ to satisfy the estimated upper bound:

$$Q_{n+1}'(\boldsymbol{s}, \boldsymbol{a}) = Q_{n+1}^{\lambda^*}(\boldsymbol{s}, \boldsymbol{a}) \wedge Q_{n+1}(\boldsymbol{s}, \boldsymbol{a}), \tag{12}$$

where $a \wedge b = \min\{a, b\}$. The agent now takes an action in the environment using a behavioral policy, such as the $\epsilon$-greedy policy on $Q'_{n+1}(\boldsymbol{s}, \boldsymbol{a})$.

The motivation behind this projection is as follows: since the subproblems are significantly smaller in terms of state and action spaces compared to the main problem, the subagents are expected to quickly converge. As a result, our upper bound estimates will get better, improving the the action-value estimate of the main Q-learning agent through the projection step. In addition, WCQL can enable a sort of "generalization" to unseen states by leveraging the weakly-coupled structure. The following example illustrates this piece of intuition.

**Example 1.** Suppose a WCMDP has $N = 3$ subproblems with $\mathcal{S}_i = \{1, 2, 3\}$ and $\mathcal{A}_i = \{1, 2\}$ for each $i$, leading to $3^3 \cdot 2^3 = 216$ total state action pairs. For the sake of illustration, suppose that the agent has visited states $\boldsymbol{s} = (1, 1, 1)$, $\boldsymbol{s} = (2, 2, 2)$, and $\boldsymbol{s} = (3, 3, 3)$ and both actions from each of these states. This means that from the perspective of every subproblem $i$, the agent has visited all state-action pairs in $\mathcal{S}_i \times \mathcal{A}_i$, which is enough information to produce an estimate of $Q_i^\lambda(s_i, a_i)$ for all $(s_i, a_i)$ and, interestingly, an estimate of $Q^{\lambda^*}(\boldsymbol{s}, \boldsymbol{a})$ for *every* $(\boldsymbol{s}, \boldsymbol{a})$, despite having visited only a small fraction $(6/216)$ of the possible state-action pairs. This allows the main Q-learning agent to make use of upper bound information at every state-action pair via the projection step (12). The main intuition is that these upper bound values are likely to be more sensible than a randomly initialized value, and therefore, can aid learning.

The above example is certainly contrived, but hopefully illustrates the benefits of decomposition and subsequent projection. We note that, especially in settings where the limiting factor is the ability to collect enough experience, one can trade-off extra computation to derive these bounds and improve RL performance without the need to collect additional experience. The full pseudo-code of the WCQL algorithm is available in Appendix B.

### 4.3 Convergence Analysis

In this section, we show that WCQL converges to $Q^*$ with probability one. First, we state a standard assumption on learning rates and state visitation.

**Assumption 2.** We assume the following: (i) $\sum_{n=0}^\infty \alpha_n(\boldsymbol{s}, \boldsymbol{a}) = \infty$, $\sum_{n=0}^\infty \alpha_n^2(\boldsymbol{s}, \boldsymbol{a}) < \infty$ for all $(\boldsymbol{s}, \boldsymbol{a}) \in \mathcal{S} \times \mathcal{A}$; (ii) analogous conditions to (i) hold for $\{\beta_n(s_i, a_i)\}_n$ and $\{\eta_n(w)\}_n$, and (iii) the behavioral policy is such that all state-action pairs $(\boldsymbol{s}, \boldsymbol{a})$ are visited infinitely often $w.p.1$.

**Theorem 1.** *Under Assumptions 1 and 2, the following statements hold with probability one.*

*(i) For each $i$ and $\lambda \in \Lambda$, $\lim_{n \to \infty} Q_{i,n}^\lambda(s_i, a_i) = Q_i^\lambda(s_i, a_i)$ for all $(s_i, a_i) \in \mathcal{S}_i \times \mathcal{A}_i$.*

*(ii) For each $\lambda \in \Lambda$, $\lim_{n \to \infty} Q_n^\lambda(\boldsymbol{s}, \boldsymbol{a}) \geq Q^*(\boldsymbol{s}, \boldsymbol{a})$ for all $(\boldsymbol{s}, \boldsymbol{a}) \in \mathcal{S} \times \mathcal{A}$.*

*(iii) $\lim_{n \to \infty} Q_n'(\boldsymbol{s}, \boldsymbol{a}) = Q^*(\boldsymbol{s}, \boldsymbol{a})$ for all $(\boldsymbol{s}, \boldsymbol{a}) \in \mathcal{S} \times \mathcal{A}$.*

Theorem 1 ensures that each subagent's value functions converge to the subproblem optimal value. Furthermore, it shows that asymptotically, the Lagrangian action-value function given by (9) will be an upper bound on the optimal action-value function $Q^*$ of the full problem and that our algorithm will converge to $Q^*$.

## 5 Weakly Coupled DQN

In this section, we propose our main algorithm *weakly coupled DQN* (WCDQN), which integrates the main idea of WCQL into a function approximation setting. WCDQN guides DQN using Lagrangian relaxation bounds, implemented using a constrained optimization approach.

**Networks.** Analogous to WCQL, WCDQN has a main network $Q'(\boldsymbol{s}, \boldsymbol{a}; \theta)$ that learns the action value of the full problem. In addition to the main network, WCDQN uses a subagent network $Q_i^\lambda(s_i, a_i; \theta_U)$ network to learn the subproblem action-value functions $Q_i^\lambda$. As in standard DQN, we also have $\theta^-$ and $\theta_U^-$, which are versions of $\theta$ and $\theta_U$ frozen from a previous iteration and used for computing target values [43]. The inputs to this network are $(i, \lambda, s_i, a_i)$, meaning that we can use a single network to learn the action-value function for *all* subproblems and $\lambda \in \Lambda$ simultaneously. The

Lagrangian upper bound and the best upper bound estimates are:

$$Q^\lambda(\boldsymbol{s},\boldsymbol{a};\theta_U^-) = \lambda^\intercal \boldsymbol{B}(w) + \sum_{i=1}^N Q_i^\lambda(s_i,a_i;\theta_U^-) \ \ \text{and} \ \ Q^{\lambda^*}(\boldsymbol{s},\boldsymbol{a};\theta_U^-) = \min_{\lambda \in \Lambda} Q^\lambda(\boldsymbol{s},\boldsymbol{a};\theta_U^-). \quad (13)$$

**Loss functions.** Before diving into the training process, we describe the loss functions used to train each network, as they are instructive toward understanding the main idea behind WCDQN (and how it differs from standard DQN). Consider a behavioral distribution $\rho$ for state-action pairs and a distribution $\mu$ over the multipliers $\Lambda$.

$$l_U(\theta_U) = \mathbb{E}_{\boldsymbol{s},\boldsymbol{a}\sim\rho,\lambda\sim\mu}\left[\sum_{i=1}^N \big(y_i^\lambda - Q_i^\lambda(s_i,a_i;\theta_U)\big)^2\right], \quad (14)$$

where the (ideal) target value is

$$y_i^\lambda = r_i(s_i,a_i) - \lambda a_i + \gamma\,\mathbb{E}\big[\max_{a_i' \in \mathcal{A}_i} Q_i^\lambda(s_i',a_i';\theta_U^-)\big]. \quad (15)$$

For the main agent, we propose a loss function that adds a soft penalty for violating the upper bound:

$$l(\theta) = \mathbb{E}_{\boldsymbol{s},\boldsymbol{a}\sim\rho,\lambda\sim\mu}\left[\big(y - Q'(\boldsymbol{s},\boldsymbol{a};\theta)\big)^2 + \kappa_U\big(Q'(\boldsymbol{s},\boldsymbol{a};\theta) - y_U\big)_+^2\right], \quad (16)$$

where $(\cdot)_+ = \max(\cdot,0)$, $\kappa_U$ is a coefficient for the soft penalty, and

$$y = r(\boldsymbol{s},\boldsymbol{a}) + \gamma\mathbb{E}\big[\max_{\boldsymbol{a}'\in\mathcal{A}(\boldsymbol{s}')} Q'(\boldsymbol{s}',\boldsymbol{a}';\theta^-)\big], \quad (17)$$

$$y_U = r(\boldsymbol{s},\boldsymbol{a}) + \gamma\mathbb{E}\big[\max_{\boldsymbol{a}'\in\mathcal{A}} Q^{\lambda^*}(\boldsymbol{s}',\boldsymbol{a}';\theta_U^-)\big]. \quad (18)$$

The penalty encourages the network to satisfy the bounds obtained from the Lagrangian relaxation.

**Training process.** The training process resembles DQN, with a few modifications. At any iteration, we first take an action using an $\epsilon$-greedy policy using the main network over the feasible actions, store the obtained transition experience $\tau$ in the buffer, and update the estimate of $\boldsymbol{B}(w)$ following (8).[2] Each network is then updated by taking a stochastic gradient descent step on its associated loss function, where the expectations are approximated by sampling minibatches of experience tuples $\tau$ and $\lambda$. The penalty coefficient $\kappa_U$ can either be held constant to a positive value or annealed using a schedule throughout the training. The full details are shown in Algorithm 1 and some further details are given in Appendix C.

## 6 Numerical Experiments

In this section, we evaluate our algorithms on three different WCMDPs. First, we evaluate WCQL on an electric vehicle (EV) deadline scheduling problem with multiple charging spots and compare its performance with several other tabular algorithms: Q-learning (QL), Double Q-learning (Double-QL) [22], speedy Q-learning (SQL) [4], bias-corrected Q-learning (BCQL) [40], and Lagrange policy Q-learning (Lagrangian QL) [34]. We then evaluate WCDQN on two problems, multi-product inventory control and online stochastic ad matching, and compare against standard DQN, Double-DQN, and the optimality-tightening DQN (OTDQN) algorithm[3] of He et al. [25] as baselines. Further details on environment and algorithmic parameters are in Appendix D.

**EV charging deadline scheduling [63].** In this problem, a decision maker is responsible for charging electric vehicles (EV) at a charging service center that consists of $N = 3$ charging spots. An EV enters the system when a charging spot is available and announces the amount of electricity it needs to be charged, denoted $B_t$, along with the time that it will leave the system, denoted $D_t$. The decision maker also faces exogenous, random Markovian processing costs $c_t$. At each period, the action is to decide which EVs to charge in accordance with the period's capacity constraint. For each unit of power provided to an EV, the service center receives a reward $1 - c_t$. However, if the EV leaves the system with an unfulfilled charge, a penalty is assessed. The goal is to maximize the revenue minus penalty costs.

---

[2]Here we use a tabular representation for $\boldsymbol{B}(w)$ since our example applications do not necessarily have a large exogenous space $\mathcal{W}$. When required, WCDQN can be extended to use function approximation (i.e., neural networks) to learn $\boldsymbol{B}(w)$.

[3]We include OTDQN as a baseline because it also makes use of constrained optimization during training.

---

**Algorithm 1** Weekly Coupled DQN

---

1: **Input**: Lagrange multiplier set $\Lambda$ and a distribution $\mu$ over $\Lambda$, penalty coefficient $\kappa_U$, initialized replay buffer $\mathcal{D}$, target network update frequency $C_{\text{target}}$, initial state distribution $S_0$.
2: Initialize main Q-network $Q'(\cdot, \cdot\,; \theta)$ and subagent network $Q_i^\lambda(\cdot, \cdot\,; \theta_U)$
3: Initialize target network weights $\theta^- = \theta$ and $\theta_U^- = \theta_U$.
4: **for** $n = 0, 1, 2, \ldots$ **do**
5:     Take an $\epsilon$-greedy behavioral action $\boldsymbol{a}_n$ with respect to the main network $Q'(\boldsymbol{s}_n, \boldsymbol{a}; \theta)$.
6:     Store the observed transition $\boldsymbol{\tau}_n$ into the replay buffer $\mathcal{D}$.
7:     `// Update subagents and combine results to estimate upper bound`
8:     Sample a minibatch of transitions $\boldsymbol{\tau}$ from $\mathcal{D}$ along with a sample of $\lambda$ from $\mu$.
9:     **for** $i = 1, 2, \ldots, N$ **do**
10:         Compute targets $y_i^\lambda$ using (15) and take an optimization step on subagent loss (14).
11:     **end for**
12:     Update right-hand-side estimate $\boldsymbol{B}_{n+1}(w_n)$ according to (8).
13:     Using (13), combine subproblems to obtain Lagrangian upper bound $Q^{\lambda^*}(\boldsymbol{s}, \boldsymbol{a}; \theta_U^-)$.
14:     `// Main agent update with upper bound penalty loss`
15:     Compute $y$ and $y_U$ using (17) and (18), then take an optimization step on the main loss (16).
16:     Update $\theta^- = \theta$ and $\theta_U^- = \theta_U$ every $C_{\text{target}}$ steps.
17: **end for**

---

**Multi-product inventory control with an exogenous production rate [27].** Consider the problem of resource allocation for a facility that manufactures $N = 10$ products. Each product $i$ has an independent exogenous demand given by $D_i$, $i = 1, \ldots, N$. To meet these demands, the products are made to stock. Limited storage $R_i$ is available for each product, and holding a unit of inventory per period incurs a cost $h_i$. Unmet demand is backordered at a cost $b_i$ if the number of backorders is less than the maximum number of allowable backorders $M_i$. Otherwise, it is lost with a penalty cost $l_i$. The DM needs to allocate a resource level $a_i \in \{0, 1, \ldots, U\}$ for product $i$ from a shared finite resource quantity $U$ in response to changes in the stock level of each product, denoted by $x_i \in X_i = \{-M_i, -M_i + 1, \ldots, R_i\}$. A negative stock level corresponds to the number of backorders. Allocating a resource level $a_i$ yields a production rate given by a function $\rho_i(a_i, p_i)$ where $p_i$ is an exogenous Markovian noise that affects the production rate. The goal is to minimize the total cost, which consists of holding, back-ordering, and lost sales costs.

**Online stochastic ad matching [18].** We study the problem of matching $N = 6$ advertisers to arriving impressions. In each period, an impression of type $e_t$ arrives according to a Markov chain. An action $a_{t,i} \in \{0, 1\}$ assigns impression $e_t$ to advertiser $i$, with a constraint that exactly one advertiser is selected: $\sum_{i=1}^{N} a_{t,i} = 1$. Advertiser states represent the number of remaining ads to display and evolves according to $s_{t+1,i} = s_{t,i} - a_{t,i}$. The objective is to maximize the discounted sum of expected rewards for all advertisers.

In Figure 2, we show how WCQL's projection method helps it learn a more accurate $Q$ function more quickly than competing tabular methods. The first panel, Figure 2A, shows an example evolution of WCQL's projected value function $Q'$, along with the evolution of the upper bound. We compare this to the evolution of the action-value function in absence of the projection step. In the second panel, Figure 2B, we plot the *relative error* between the learned value functions of various algorithms compared to the optimal value function. Both plots are from the EV charging example. Detailed descriptions of the results are given in the figure's caption.

The results of our numerical experiments are shown in Figure 3. We see that in both the tabular and the function approximation cases, our algorithms outperformed the baselines, with WCQL and WCDQN achieving the best mean episode total rewards amongst all problems. From Figure 3A, we see that although the difference between WCQL and Lagrangian QL is small towards the end of the training process, there are stark differences earlier on. In particular, the performance curve of WCQL shows significantly lower variance, suggesting more robustness across random seeds. Given that WCQL and Lagrangian QL differ only in the projection step, we can attribute the improved stability to the guidance provided by the Lagrangian bounds. Figure 3B shows that for the multi-product inventory problem, the OTDQN, DQN, and Double DQN baselines show extremely noisy performance compared to WCDQN, whose significantly better and stable performance is likely

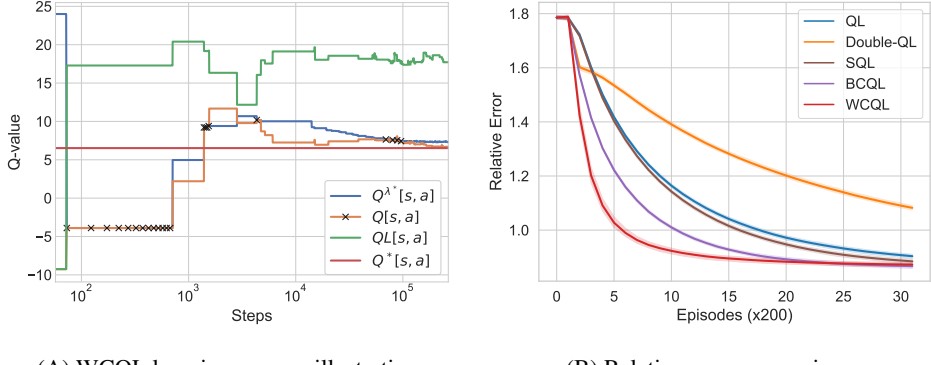

(A) WCQL learning process illustration

(B) Relative error comparison

Figure 2: Behavior of WCQL's learning process compared to other tabular methods. In Figure 2A, we show an example of the evolution of quantities associated with WCQL for a randomly selected state-action pair in the EV charging problem: upper bound (blue), WCQL Q-value (orange line with 'x' marks indicating the points projected by the bound), standard Q-learning (green) and the optimal action-value (red). Note that sometimes the orange line appears above the blue line since WCQL's projection step is asynchronous, i.e., the projections are made only if the state is visited by the behavioral policy. Notice at the last marker, the bound moved the Q-value in orange to a "good" value, relatively nearby the optimal value. WCQL's Q-value then evolves on its own and eventually converges. On the other hand, standard QL (green), which follows the same behavior policy as WCQL, is still far from optimal. In Figure 2B, we show the relative error, defined as $\|V_n - V^*\|_2/\|V^*\|_2$, where $V_n$ and $V^*$ are the state value functions derived from $Q$-iterate on iteration $n$ and $Q^*$, respectively. WCQL exhibits the steepest decline in relative error compared to the other algorithms. Note that since Lagrangian QL acts based on the Q-values of the subproblems, there is no Q-value for this algorithm on which to compute a relative error.

due to the use of faster converging subagents and better use of the collected experience. Similarly, in the online stochastic ad matching problem, WCDQN significantly outperforms all the baselines.

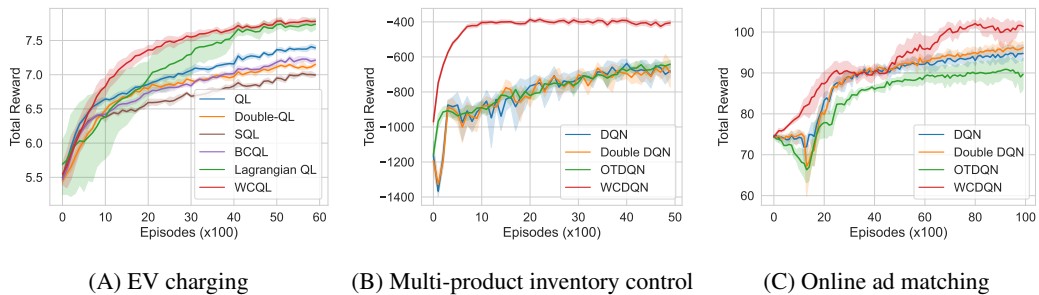

(A) EV charging

(B) Multi-product inventory control

(C) Online ad matching

Figure 3: Benchmarking results for the WCQL (EV charging) and WCDQN (multi-product inventory control, online ad matching) against baseline methods. The plots show mean total rewards and their 95% confidence intervals across 5 independent replications.

## 7 Conclusion

In this study, we propose the WCQL algorithm for learning in weakly coupled MDPs and we show that our algorithm converges to the optimal action-value function. We then propose WCDQN, which extends the idea behind the WCQL algorithm to the function approximation case. Our algorithms are model-free and learn upper bounds on the optimal action-value using a combination of a Lagrangian relaxation and Q-learning. These bounds are then used within a constrained optimization approach to improve performance and make learning more efficient. Our approaches significantly outperforms competing approaches on several benchmark environments.

## Acknowledgments and Disclosure of Funding

This research was supported in part by the University of Pittsburgh Center for Research Computing, RRID:SCR_022735, through the resources provided. Specifically, this work used the H2P cluster, which is supported by NSF award number OAC-2117681.

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

# Appendix to *Weakly Coupled Deep Q-Networks*

## A Proofs

### A.1 Proof of Proposition 1

*Proof.* We prove part the first part of the proposition (weak duality) by induction. First, define

$$Q_0^*(\boldsymbol{s}, \boldsymbol{a}) = r(\boldsymbol{s}, \boldsymbol{a}) \quad \text{and} \quad Q_0^\lambda(\boldsymbol{s}, \boldsymbol{a}) = r(\boldsymbol{s}, \boldsymbol{a}) + \lambda^\intercal \left[ \boldsymbol{b}(w) - \sum_{i=1}^N \boldsymbol{d}(s_i, a_i) \right],$$

and suppose we run value iteration for both systems:

$$Q_{t+1}^*(\boldsymbol{s}, \boldsymbol{a}) = r(\boldsymbol{s}, \boldsymbol{a}) + \gamma \, \mathbb{E}\big[\max_{\boldsymbol{a}' \in \mathcal{A}(\boldsymbol{s}')} Q_t^*(\boldsymbol{s}', \boldsymbol{a}')\big],$$

$$Q_{t+1}^\lambda(\boldsymbol{s}, \boldsymbol{a}) = r(\boldsymbol{s}, \boldsymbol{a}) + \lambda^\intercal \left[ \boldsymbol{b}(w) - \sum_{i=1}^N \boldsymbol{d}(s_i, a_i) \right] + \gamma \, \mathbb{E}\big[\max_{\boldsymbol{a}' \in \mathcal{A}} Q_t^\lambda(\boldsymbol{s}', \boldsymbol{a}')\big].$$

It is well-known that, by the value iteration algorithm's convergence,

$$Q^*(\boldsymbol{s}, \boldsymbol{a}) = \lim_{t \to \infty} Q_t^*(\boldsymbol{s}, \boldsymbol{a}) \quad \text{and} \quad Q^\lambda(\boldsymbol{s}, \boldsymbol{a}) = \lim_{t \to \infty} Q_t^\lambda(\boldsymbol{s}, \boldsymbol{a}).$$

Consider a state $\boldsymbol{s} \in \mathcal{S}$ and a feasible action $\boldsymbol{a} \in \mathcal{A}(\boldsymbol{s})$. We have,

$$Q_0^\lambda(\boldsymbol{s}, \boldsymbol{a}) = r(\boldsymbol{s}, \boldsymbol{a}) + \lambda^\intercal \left[ \boldsymbol{b}(w) - \sum_{i=1}^N \boldsymbol{d}(s_i, a_i) \right] \geq r(\boldsymbol{s}, \boldsymbol{a}) = Q_0^*(\boldsymbol{s}, \boldsymbol{a}).$$

Suppose $Q_t^\lambda(\boldsymbol{s}, \boldsymbol{a}) \geq Q_t^*(\boldsymbol{s}, \boldsymbol{a})$ holds for all $\boldsymbol{s} \in \mathcal{S}$ and $\boldsymbol{a} \in \mathcal{A}(\boldsymbol{s})$ for some $t > 0$ (induction hypothesis). Then,

$$Q_{t+1}^\lambda(\boldsymbol{s}, \boldsymbol{a}) = r(\boldsymbol{s}, \boldsymbol{a}) + \lambda^\intercal \left[ \boldsymbol{b}(w) - \sum_{i=1}^N \boldsymbol{d}(s_i, a_i) \right] + \gamma \, \mathbb{E}\big[\max_{\boldsymbol{a}' \in \mathcal{A}} Q_t^\lambda(\boldsymbol{s}', \boldsymbol{a}')\big]$$

$$\geq r(\boldsymbol{s}, \boldsymbol{a}) + \lambda^\intercal \left[ \boldsymbol{b}(w) - \sum_{i=1}^N \boldsymbol{d}(s_i, a_i) \right] + \gamma \, \mathbb{E}\big[\max_{\boldsymbol{a}' \in \mathcal{A}(\boldsymbol{s}')} Q_t^*(\boldsymbol{s}', \boldsymbol{a}')\big]$$

$$\geq r(\boldsymbol{s}, \boldsymbol{a}) + \gamma \, \mathbb{E}\big[\max_{\boldsymbol{a}' \in \mathcal{A}(\boldsymbol{s}')} Q_t^*(\boldsymbol{s}', \boldsymbol{a}')\big] = Q_{t+1}^*(\boldsymbol{s}, \boldsymbol{a}).$$

Thus, it follows that $Q^\lambda(\boldsymbol{s}, \boldsymbol{a}) \geq Q^*(\boldsymbol{s}, \boldsymbol{a})$.

For the proof of the second part of the proposition, define

$$\boldsymbol{B}_0(w) = \boldsymbol{b}(w) \quad \text{and} \quad \boldsymbol{B}_{t+1}(w) = \boldsymbol{b}(w) + \gamma \, \mathbb{E}\big[\boldsymbol{B}_t(w')\big].$$

We use an induction proof. We have for all $(\boldsymbol{s}, \boldsymbol{a}) \in \mathcal{S} \times \mathcal{A}$,

$$Q_0^\lambda(\boldsymbol{s}, \boldsymbol{a}) = r(\boldsymbol{s}, \boldsymbol{a}) + \lambda^\intercal \left[ \boldsymbol{b}(w) - \sum_{i=1}^N \boldsymbol{d}(s_i, a_i) \right]$$

$$= \sum_{i=1}^N \Big[ r_i(s_i, a_i) - \lambda^\intercal \boldsymbol{d}(s_i, a_i) \Big] + \lambda^\intercal \boldsymbol{b}(w) = \sum_{i=1}^N Q_{0,i}^\lambda(s_i, a_i) + \lambda^\intercal \boldsymbol{B}_0(w),$$

where $Q_{0,i}^\lambda(s_i, a_i) = r_i(s_i, a_i) - \lambda^\intercal \boldsymbol{d}(s_i, a_i)$. Similarly, for all $(\boldsymbol{s}, \boldsymbol{a}) \in \mathcal{S} \times \mathcal{A}$,

$$Q_1^\lambda(\boldsymbol{s}, \boldsymbol{a}) = r(\boldsymbol{s}, \boldsymbol{a}) + \lambda^\intercal \left[ \boldsymbol{b}(w) - \sum_{i=1}^N \boldsymbol{d}(s_i, a_i) \right] + \gamma \, \mathbb{E}\big[\max_{\boldsymbol{a}' \in \mathcal{A}} Q_0^\lambda(\boldsymbol{s}', \boldsymbol{a}')\big]$$

$$= \sum_{i=1}^N \Big[ r_i(s_i, a_i) - \lambda^\intercal \boldsymbol{d}(s_i, a_i) \Big] + \lambda^\intercal \boldsymbol{b}(w) + \gamma \, \mathbb{E}\left[ \max_{\boldsymbol{a}' \in \mathcal{A}} \left\{ \sum_{i=1}^N Q_{0,i}^\lambda(s_i', a_i') + \lambda^\intercal \boldsymbol{B}_0(w') \right\} \right]$$

$$= \sum_{i=1}^N \Big[ r_i(s_i, a_i) - \lambda^\intercal \boldsymbol{d}(s_i, a_i) + \gamma \, \mathbb{E}\big[\max_{a_i' \in \mathcal{A}_i} Q_{0,i}^\lambda(s_i', a_i')\big] \Big] + \lambda^\intercal \Big( \boldsymbol{b}(w) + \gamma \, \mathbb{E}\big[\boldsymbol{B}_0(w')\big] \Big)$$

$$= \sum_{i=1}^N Q_{1,i}^\lambda(s_i, a_i) + \lambda^\intercal \boldsymbol{B}_1(w).$$

Continuing in this manner, we arrive at $Q_t^\lambda(\boldsymbol{s}, \boldsymbol{a}) = \sum_{i=1}^N Q_{t,i}^\lambda(s_i, a_i) + \lambda^\intercal \boldsymbol{B}_t(w)$. Finally, we have

$$
\begin{aligned}
Q^\lambda(\boldsymbol{s}, \boldsymbol{a}) &= \lim_{t \to \infty} Q_t^\lambda(\boldsymbol{s}, \boldsymbol{a}) \\
&= \lim_{t \to \infty} \sum_{i=1}^N Q_{t,i}^\lambda(s_i, a_i) + \lambda^\intercal \boldsymbol{B}_t(w) = \sum_{i=1}^N Q_i^\lambda(s_i, a_i) + \lambda^\intercal \boldsymbol{B}(w),
\end{aligned}
$$

which follows by the convergence of value iteration. $\qquad\square$

## A.2 Proof of Theorem 1

*Proof.* First, we define the Bellman operator $H$:

$$
(HQ')(\boldsymbol{s}, \boldsymbol{a}) = r(\boldsymbol{s}, \boldsymbol{a}) + \gamma \mathbb{E}\left[\max_{\boldsymbol{a}'} Q'(\boldsymbol{s}', \boldsymbol{a}')\right],
$$

which is known to be a $\gamma$-contraction mapping. Next we define the random noise term

$$
\xi_n(\boldsymbol{s}, \boldsymbol{a}) = \gamma \max_{\boldsymbol{a}'} Q_n'(\boldsymbol{s}', \boldsymbol{a}') - \gamma \mathbb{E}\left[\max_{\boldsymbol{a}'} Q_n'(\boldsymbol{s}', \boldsymbol{a}')\right]. \tag{19}
$$

Analogously, for subproblem $i \in \{1, \dots, N\}$, define the subproblem Bellman operator

$$
(H_i Q_i^\lambda)(s_i, a_i) = r_i(s_i, a_i) - \lambda^\intercal \boldsymbol{d}(s_i, a_i) + \gamma \mathbb{E}\left[\max_{a_i'} Q_i^\lambda(s_i', a_i')\right],
$$

and random noise term

$$
\xi_{i,n}(s_i, a_i) = \gamma \max_{a_i'} Q_{i,n}'(s_i', a_i') - \gamma \mathbb{E}\left[\max_{a_i'} Q_{i,n}'(s_i', a_i')\right]. \tag{20}
$$

The update rules of WCQL can then be written as

$$
\begin{aligned}
Q_{n+1}(\boldsymbol{s}, \boldsymbol{a}) &= (1 - \alpha_n(\boldsymbol{s}, \boldsymbol{a})) Q_n'(\boldsymbol{s}, \boldsymbol{a}) + \alpha_n(\boldsymbol{s}, \boldsymbol{a}) \left[(HQ_n')(\boldsymbol{s}, \boldsymbol{a}) + \xi_{n+1}(\boldsymbol{s}, \boldsymbol{a})\right], \\
Q_{i,n+1}^\lambda(s_i, a_i) &= Q_{i,n}^\lambda(s_i, a_i) + \beta_n(s_i, a_i) \left[(H_i Q_{i,n}')(s_i, a_i) + \xi_{i,n+1}(s_i, a_i)\right], \tag{21}
\end{aligned}
$$

$$
\begin{aligned}
Q_{n+1}^{\lambda^*}(\boldsymbol{s}, \boldsymbol{a}) &= \min_{\lambda \in \Lambda} \lambda^\intercal \boldsymbol{B}_n(w) + \sum_{i=1}^N Q_{i,n}^\lambda(s_i, a_i), \\
Q_{n+1}'(\boldsymbol{s}, \boldsymbol{a}) &= \min(Q_{n+1}(\boldsymbol{s}, \boldsymbol{a}), Q_{n+1}^{\lambda^*}(\boldsymbol{s}, \boldsymbol{a})). \tag{22}
\end{aligned}
$$

**Parts (i) and (ii).** By the iteration described in (21), we know that for a fixed $\lambda$, we are running Q-learning on an auxiliary MDP with Bellman operator $H_i$, which encodes a reward $r_i(s_i, a_i) - \lambda^\intercal \boldsymbol{d}(s_i, a_i)$ and the transition dynamics for subproblem $i$. By the standard result for asymptotic convergence of Q-learning [6], we have

$$
\lim_{n \to \infty} Q_{i,n}^\lambda(s_i, a_i) = Q_i^\lambda(s_i, a_i). \tag{23}
$$

We now prove the result in (ii): $\lim_{n \to \infty} Q_n^\lambda(\boldsymbol{s}, \boldsymbol{a}) \geq Q^*(\boldsymbol{s}, \boldsymbol{a})$. Recall that

$$
Q_n^\lambda(\boldsymbol{s}, \boldsymbol{a}) = \lambda^\intercal \boldsymbol{B}_n(w) + \sum_{i=1}^N Q_{i,n}^\lambda(s_i, a_i).
$$

By standard stochastic approximation theory, $\lim_{n \to \infty} \boldsymbol{B}_n(w) = \boldsymbol{B}(w)$ for all $w$ [38]. Combining this with (23), we have $\lim_{n \to \infty} Q_n^\lambda(\boldsymbol{s}, \boldsymbol{a}) = Q^\lambda(\boldsymbol{s}, \boldsymbol{a})$ for all $(\boldsymbol{s}, \boldsymbol{a})$, and to conclude that this limit is an upper bound on $Q^*(\boldsymbol{s}, \boldsymbol{a})$, we apply Proposition 1.

**Part (iii).** Assume without loss of generality that $Q^*(\boldsymbol{s}, \boldsymbol{a}) = 0$ for all state-action pairs $(\boldsymbol{s}, \boldsymbol{a})$. This can be established by shifting the origin of the coordinate system. We also assume that $\alpha_n(\boldsymbol{s}, \boldsymbol{a}) \leq 1$ for all $(\boldsymbol{s}, \boldsymbol{a})$ and $n$. We proceed via induction. Note that the iterates $Q_n'(\boldsymbol{s}, \boldsymbol{a})$ are bounded in the sense that there exists a constant $D_0 = R_{\max}/(1 - \gamma)$, $R_{\max} = \max_{(\boldsymbol{s}, \boldsymbol{a})} |r(\boldsymbol{s}, \boldsymbol{a})|$, such that $|Q_n'(\boldsymbol{s}, \boldsymbol{a})| \leq D_0$ for all $(\boldsymbol{s}, \boldsymbol{a})$ and iterations $n$ [17]. Define the sequence $D_{k+1} = (\gamma + \epsilon) D_k$, such that $\gamma + \epsilon < 1$ and $\epsilon > 0$. Clearly, $D_k \to 0$. Suppose that there exists a random variable $n_k$, representing an iteration threshold such that for all $(\boldsymbol{s}, \boldsymbol{a})$,

$$
-D_k \leq Q_n'(\boldsymbol{s}, \boldsymbol{a}) \leq \min\{D_k, Q_n^{\lambda^*}(\boldsymbol{s}, \boldsymbol{a})\}, \quad \forall n \geq n_k.
$$

We will show that there exists some iteration $n_{k+1}$ such that

$$-D_{k+1} \leq Q'_n(\boldsymbol{s}, \boldsymbol{a}) \leq \min\{D_{k+1}, Q_n^{\lambda^*}(\boldsymbol{s}, \boldsymbol{a})\} \quad \forall (s, a), \, n \geq n_{k+1},$$

which implies that $Q'_n(\boldsymbol{s}, \boldsymbol{a})$ converges to $Q^*(\boldsymbol{s}, \boldsymbol{a}) = 0$ for all $(\boldsymbol{s}, \boldsymbol{a})$.

By part (ii), we know that for all $\eta > 0$, with probability 1, there exists some finite iteration $n_0$ such that for all $n \geq n_0$,

$$Q^*(\boldsymbol{s}, \boldsymbol{a}) - \eta \leq Q_n^{\lambda^*}(\boldsymbol{s}, \boldsymbol{a}). \tag{24}$$

Now, we define an accumulated noise process started at $n_k$ by $W_{n_k, n_k}(\boldsymbol{s}, \boldsymbol{a}) = 0$, and

$$W_{n+1, n_k}(\boldsymbol{s}, \boldsymbol{a}) = (1 - \alpha_n(\boldsymbol{s}, \boldsymbol{a})) W_{n, n_k}(\boldsymbol{s}, \boldsymbol{a}) + \alpha_n(\boldsymbol{s}, \boldsymbol{a}) \xi_{n+1}(\boldsymbol{s}, \boldsymbol{a}), \quad \forall n \geq n_k, \tag{25}$$

where $\xi_n(\boldsymbol{s}, \boldsymbol{a})$ is as defined in (19). Let $\mathcal{F}_n$ be the entire history of the algorithm up to the point where the step sizes at iteration $n$ are selected. Using Corollary 4.1 in [6] which states that under Assumption 2 on the step size $\alpha_n(\boldsymbol{s}, \boldsymbol{a})$, and if $\mathbb{E}[\xi_n(\boldsymbol{s}, \boldsymbol{a}) \,|\, \mathcal{F}_n] = 0$ and $\mathbb{E}[\xi_n^2(\boldsymbol{s}, \boldsymbol{a}) \,|\, \mathcal{F}_n] \leq A_n$, where the random variable $A_n$ is bounded with probability 1, the sequence $W_{n+1, n_k}(\boldsymbol{s}, \boldsymbol{a})$ defined in (25) converges to zero, with probability 1. From our definition of the stochastic approximation noise $\xi_n(\boldsymbol{s}, \boldsymbol{a})$ in (19), we have

$$\mathbb{E}[\xi_n(\boldsymbol{s}, \boldsymbol{a}) \,|\, \mathcal{F}_n] = 0 \quad \text{and} \quad \mathbb{E}[\xi_n^2(\boldsymbol{s}, \boldsymbol{a}) \,|\, \mathcal{F}_n] \leq C(1 + \max_{\boldsymbol{s}', \boldsymbol{a}'} Q_n'^2(\boldsymbol{s}', \boldsymbol{a}')),$$

where $C$ is a constant. Then, it follows that

$$\lim_{n \to \infty} W_{n, n_k}(\boldsymbol{s}, \boldsymbol{a}) = 0 \quad \forall (\boldsymbol{s}, \boldsymbol{a}), \, n_k.$$

We use the following lemma from [6] to bound the accumulated noise.

**Lemma A.1** (Lemma 4.2 in [6]). *For every $\delta > 0$, with probability one, there exists some $n'$ such that $|W_{n, n'}(\boldsymbol{s}, \boldsymbol{a})| \leq \delta$, for all $n \geq n'$.*

Now, by Lemma A.1, let $n_{k'} \geq \max(n_k, n_0)$ such that, for all $n \geq n_{k'}$ we have

$$|W_{n, n_{k'}}(\boldsymbol{s}, \boldsymbol{a})| \leq \gamma \epsilon D_k < \gamma D_k.$$

Let $\nu_k \geq n_{k'}$ such that, for all $n \geq \nu_k$, by (24) we have

$$\gamma \epsilon D_k - \gamma D_k \leq Q_n^{\lambda^*}(\boldsymbol{s}, \boldsymbol{a}).$$

Define another sequence $Y_n$ that starts at iteration $\nu_k$.

$$Y_{\nu_k}(\boldsymbol{s}, \boldsymbol{a}) = D_k \quad \text{and} \quad Y_{n+1}(\boldsymbol{s}, \boldsymbol{a}) = (1 - \alpha_n(\boldsymbol{s}, \boldsymbol{a})) Y_n(\boldsymbol{s}, \boldsymbol{a}) + \alpha_n(\boldsymbol{s}, \boldsymbol{a}) \gamma D_k \tag{26}$$

Note that it is easy to show that the sequence $Y_n(\boldsymbol{s}, \boldsymbol{a})$ in (26) is decreasing, bounded below by $\gamma D_k$, and converges to $\gamma D_k$ as $n \to \infty$. Now we state the following lemma.

**Lemma A.2.** *For all state-action pairs $(\boldsymbol{s}, \boldsymbol{a})$ and iterations $n \geq \nu_k$, it holds that:*

$$-Y_n(\boldsymbol{s}, \boldsymbol{a}) + W_{n, \nu_k}(\boldsymbol{s}, \boldsymbol{a}) \leq Q'_n(\boldsymbol{s}, \boldsymbol{a}) \leq \min\{Q_n^{\lambda^*}(\boldsymbol{s}, \boldsymbol{a}), Y_n(\boldsymbol{s}, \boldsymbol{a}) + W_{n, \nu_k}(\boldsymbol{s}, \boldsymbol{a})\}. \tag{27}$$

*Proof.* We focus on the right hand side inequality, the left hand side can be proved similarly. For the base case $n = \nu_k$, the statement holds because $Y_{\nu_k}(\boldsymbol{s}, \boldsymbol{a}) = D_k$ and $W_{\nu_k, \nu_k}(\boldsymbol{s}, \boldsymbol{a}) = 0$. We assume it is true for $n$ and show that it continues to hold for $n + 1$:

$$\begin{aligned}
Q_{n+1}(\boldsymbol{s}, \boldsymbol{a}) &= (1 - \alpha_n(\boldsymbol{s}, \boldsymbol{a})) Q'_n(\boldsymbol{s}, \boldsymbol{a}) + \alpha_n(\boldsymbol{s}, \boldsymbol{a}) \left[ (HQ'_n)(\boldsymbol{s}, \boldsymbol{a}) + \xi_{n+1}(\boldsymbol{s}, \boldsymbol{a}) \right] \\
&\leq (1 - \alpha_n(\boldsymbol{s}, \boldsymbol{a})) \min\{Q_n^{\lambda^*}(\boldsymbol{s}, \boldsymbol{a}), Y_n(\boldsymbol{s}, \boldsymbol{a}) + W_{n, \nu_k}(\boldsymbol{s}, \boldsymbol{a})\} \\
&\qquad + \alpha_n(\boldsymbol{s}, \boldsymbol{a}) (HQ'_n)(\boldsymbol{s}, \boldsymbol{a}) + \alpha_n(\boldsymbol{s}, \boldsymbol{a}) \xi_{n+1}(\boldsymbol{s}, \boldsymbol{a}) \\
&\leq (1 - \alpha_n(\boldsymbol{s}, \boldsymbol{a})) (Y_n(\boldsymbol{s}, \boldsymbol{a}) + W_{n, \nu_k}(\boldsymbol{s}, \boldsymbol{a})) + \alpha_n(\boldsymbol{s}, \boldsymbol{a}) \gamma D_k + \alpha_n(\boldsymbol{s}, \boldsymbol{a}) \xi_{n+1}(\boldsymbol{s}, \boldsymbol{a}) \\
&\leq Y_{n+1}(\boldsymbol{s}, \boldsymbol{a}) + W_{n+1, \nu_k}(\boldsymbol{s}, \boldsymbol{a}),
\end{aligned}$$

where we used $(HQ'_n) \leq \gamma \|Q'_n\| \leq \gamma D_k$. Now, we have

$$\begin{aligned}
Q'_{n+1}(\boldsymbol{s}, \boldsymbol{a}) &= \min(Q_{n+1}^{\lambda^*}(\boldsymbol{s}, \boldsymbol{a}), Q_{n+1}(\boldsymbol{s}, \boldsymbol{a})) \\
&\leq \min\{Q_{n+1}^{\lambda^*}(\boldsymbol{s}, \boldsymbol{a}), Y_{n+1}(\boldsymbol{s}, \boldsymbol{a}) + W_{n+1, \nu_k}(\boldsymbol{s}, \boldsymbol{a})\}.
\end{aligned}$$

The inequality holds because

$$Q_{n+1}(\boldsymbol{s}, \boldsymbol{a}) \leq Y_{n+1}(\boldsymbol{s}, \boldsymbol{a}) + W_{n+1, \nu_k}(\boldsymbol{s}, \boldsymbol{a}),$$

which completes the proof. $\qquad \square$

Since $Y_n(s, a) \to \gamma D_k$ and $W_{n,\nu_k}(s, a) \to 0$, we have

$$\limsup_{n\to\infty} \|Q'_n\| \leq \gamma D_k < D_{k+1}.$$

Therefore, there exists some time $n_{k+1}$ such that

$$-D_{k+1} \leq Q'_n(s, a) \leq \min\{D_{k+1}, Q_n^{\lambda^*}(s, a)\} \ \forall (s, a), \ n \geq n_{k+1},$$

which completes the induction.

$\square$

## B  Weakly Coupled Q-learning Algorithm Description

---
**Algorithm 2** Weekly Coupled Q-learning
---
1: **Input**: Lagrange multiplier set $\Lambda$, initial state distribution $S_0$.
2: Initialize Q-table estimates $Q_0, \{Q_{0,i}\}_{i=1}^N$. Set $Q'_0 = Q_0$.
3: **for** $n = 0, 1, 2, \ldots$ **do**
4:     Take an $\epsilon$-greedy behavioral action $a_n$ with respect to $Q'_n(s_n, a)$.
5:     `// Estimate upper bound by combining subagents`
6:     **for** $i = 1, 2, \ldots, N$ **do**
7:         Update each subproblem value functions $Q_{i,n+1}^\lambda$ according to (7).
8:     **end for**
9:     Update right-hand-side estimate $B_{n+1}(w_n)$ according to (8).
10:    Using (9) and (10), combine subproblems to obtain $Q_{n+1}^{\lambda^*}(s_n, a)$ for all $a \in \mathcal{A}(s_n)$.
11:    `// Main agent standard update, followed by projection`
12:    Do standard Q-learning update using (11) to obtain $Q_{n+1}$.
13:    Perform upper bound projection step: $Q'_{n+1}(s, a) = Q_{n+1}^{\lambda^*}(s, a) \wedge Q_{n+1}(s, a)$
14: **end for**
---

## C  Weakly Coupled DQN Algorithm Implementation

In our implementation of WCDQN, the subproblem $Q_i^\lambda$-network in Algorithm 1 follows the standard network architecture as in [43], where given an input state $s_i$ the network predicts the $Q$-values for all actions. This mandates that all the subproblems have the same number of actions. To address different subproblem action spaces, we can change the network architecture to receive the state-action pair $(s_i, a_i)$ as input and output the predicted $Q$-value. This simple change does not interfere or affect WCDQN's main idea.

Our code is available at https://github.com/ibrahim-elshar/WCDQN_NeurIPS.

## D  Numerical Experiment Details

A discount factor of $0.9$ is used for the EV charging problem and $0.99$ for the multi-product inventory and online stochastic ad matching problems. In the tabular setting, we use a polynomial learning rate that depends on the state-action pairs visitation given by $\alpha_n(s, a) = 1/\nu_n(s, a)^r$, where $\nu_n(s, a)$ represent the number of times $(s, a)$ has been visited up to iteration $n$, and $r = 0.4$. We also use an $\epsilon$-greedy exploration policy, given by $\epsilon(s) = 1/\nu(s)^e$, where $\nu(s)$ is the number of times the state $s$ has been visited. We set $e = 0.4$. In the function approximation setting, we use an $\epsilon$-greedy policy that decays $\epsilon$ from 1 to $0.05$ after $30,000$ steps. All state-action value functions are initialized randomly. Experiments were ran on a shared memory cluster with dual 12-core Skylake CPU (Intel Xeon Gold 6126 2.60 GHz) and 192 GB RAM/node.

### D.1  EV charging deadline scheduling [63]

In this problem, there are in total three charging spots $N = 3$. Each spot represents a subproblem with state $(c_t, B_{t,i}, D_{t,i})$, where $c_t \in \{0.2, 0.5, 0.8\}$ is the exogenous electric cost, $B_{t,i} \leq 2$ is the

amount of charge required and $D_{t,i} \leq 4$ is the remaining time until the EV leaves the system. The state space size is 36 for each subproblem. At a given period $t$, the action of each subproblem is whether to charge an EV occupying the charging spot $a_{t,i} = 1$ or not $a_{t,i} = 0$. A feasible action is given by $\sum_{i=1}^{N} a_{t,i} \leq b(c_t)$, where $b(0.2) = 3, b(0.5) = 2$, and $b(0.8) = 1$. The reward of each subproblem is given by

$$r_i\big((c_t, B_{t,i}, D_{t,i}), a_{t,i}\big) = \begin{cases} (1 - c_t)\, a_{t,i} & \text{if } B_{t,i} > 0, D_{t,i} > 1, \\ (1 - c_t)\, a_{t,i} - F(B_{t,i} - a_{t,i}) & \text{if } B_{t,i} > 0, D_{t,i} = 1, \\ 0, & \text{otherwise}, \end{cases}$$

where $F(B_{t,i} - a_{t,i}) = 0.2\,(B_{t,i} - a_{t,i})^2$ is a penalty function for failing to complete charging the EV before the deadline. The endogenous state of each subproblem evolves such that $(B_{t+1,i}, D_{t+1,i}) = (B_{t,i} - a_{t,i}, D_{t,i} - 1)$ if $D_{t,i} > 1$, and $(B_{t+1,i}, D_{t+1,i}) = (B, D)$ with probability $q(D, B)$ if $D_{t,i} \leq 1$, where $q(0,0) = 0.3$ and $q(B, D) = 0.7/11$ for all $B > 0$ and $D > 0$. The exogenous state $c_t$ evolves following the transition probabilities given by:

$$q(c_{t+1} \mid c_t) = \begin{pmatrix} 0.4 & 0.3 & 0.3 \\ 0.2 & 0.5 & 0.3 \\ 0.6 & 0.2 & 0.2 \end{pmatrix}.$$

## D.2 Multi-product inventory control with an exogenous production rate [27]

We consider manufacturing $N = 10$ products. The exogenous demand $D_{t,i}$ for each product $i \in \{1, 2, \ldots, 10\}$ follows a Poisson distribution with mean value $\mu_i$. The maximum storage capacity and the maximum number of allowable backorders (after which lost sales costs incur) for product $i$ are given by $R_i$ and $M_i$, respectively.

The state for subproblem $i$ is given by $(x_{t,i}, p_t)$, where $x_{t,i} \in X_i = \{-M_i, -M_i + 1, \ldots, R_i\}$ is the inventory level for product $i$, and $p_t$ is an exogenous and Markovian noise with support $[0.8, 1]$. A negative stock level corresponds to the number of backorders. For subproblem $i$, the action $a_{t,i}$ is the number of resources allocated to the product $i$. The maximum number of resources available for all products is $U = 3$, so feasible actions must satisfy $\sum_i a_{t,i} \leq 3$.

Allocating a resource level $a_{t,i}$ yields a production rate $\rho_i(a_{t,i}, p_t) = (12\, p_t\, a_{t,i})/(5.971 + a_{t,i})$. The cost function for product $i$ is $c_i(p_t, x_{t,i}, a_{t,i})$ and represents the sum of the holding, backorders, and lost sales costs. We let $h_i$, $b_i$, and $l_i$ denote the per-unit holding, backorder, and lost sale costs, respectively. The cost function $c_i(x_{t,i}, p_t, a_{t,i})$ is given by,

$$c_i(x_{t,i}, p_t, a_{t,i}) = h_i(x_{t,i} + \rho_i(a_{t,i}, p_t))_+ + b_i(-x_{t,i} - \rho_i(a_{t,i}, p_t))_+ \\ + l_i((D_{t,i} - x_{t,i} - \rho_i(a_{t,i}, p_t))_+ - M_i)_+,$$

where $(.)_+ = \max(., 0)$. We summarize the cost parameters and the mean demand for each product in Table 1. Finally, the transition for the inventory state of subproblem $i$ is given by

$$x_{t+1,i} = \max\big(\min(x_{t,i} + \rho_i(a_{t,i}, p_t) - D_{t,i}, R_i), -M_i\big),$$

where the exogenous noise $p_t$ evolves according to a transition matrix sampled from a Dirichlet distribution whose parameters are each sampled (once per replication) from a Uniform$(1, 5)$ distribution.

Table 1: Multi-product inventory environment parameters

| Product $i$ | 1 | 2 | 3 | 4 | 5 | 6 | 7 | 8 | 9 | 10 |
|---|---|---|---|---|---|---|---|---|---|---|
| Storage capacity $R_i$ | 20 | 30 | 10 | 15 | 10 | 10 | 25 | 30 | 15 | 10 |
| Maximum backorders $M_i$ | 5 | 5 | 5 | 5 | 5 | 5 | 5 | 5 | 5 | 5 |
| Mean demand $\mu_i$ | 0.3 | 0.7 | 0.5 | 1.0 | 1.4 | 0.9 | 1.1 | 1.2 | 0.3 | 0.6 |
| Holding cost $h_i$ | 0.1 | 0.2 | 0.05 | 0.3 | 0.2 | 0.5 | 0.3 | 0.4 | 0.15 | 0.12 |
| Backorder cost $b_i$ | 3.0 | 1.2 | 5.15 | 1.3 | 1.1 | 1.1 | 10.3 | 1.05 | 1. | 3.1 |
| Lost sales cost $l_i$ | 30.1 | 3.3 | 10.05 | 3.9 | 3.7 | 3.6 | 40.3 | 4.5 | 12.55 | 44.1 |

### D.3 Online stochastic ad matching [18]

In this problem, a platform needs to match $N = 6$ advertisers to arriving impressions [18]. An impression $e_t \in E = \{1, 2, \ldots, 5\}$ arrives according to a discrete time Markov chain with transition probabilities given by $q(e_{t+1} \mid e_t)$, where each row of the transition matrix $q$ is sampled from a Dirichlet distribution whose parameters are sampled (once per replication) from Uniform$(1, 20)$.

The action $a_{t,i} \in \{0, 1\}$ is whether to assign impression $e_t$ to advertiser $i$ or not. The platform can assign an impression to at most one advertiser: $\sum_{i=1}^{N} a_{i,t} = 1$.

The state of advertiser $i$, $x_{i,t}$ gives the number of remaining ads to display and evolves according to $x_{t+1,i} = x_{t,i} - a_{t,i}$. The initial state is $x_0 = (10, 11, 12, 10, 14, 9)$. The reward obtained from advertiser $i$ in state $s_{t,i} = (x_{t,i}, e_t)$ is $r_i(s_{t,i}, a_{t,i}) = l_{i,e_t} \min(x_{t,i}, a_{t,i})$, where the parameters $l_{i,e_t}$ are sampled (once per replication) from Uniform$(1, 4)$.

### D.4 Training parameters

Each method was trained for 6,000 episodes for the EV charging problem, 5,000 for the multi-product inventory control problem, and 10,000 episodes for the online stochastic ad matching problem. The episode lengths for the EV charging, online ad stochastic ad matching, and multi-product inventory control problems are $50, 30$, and $25$, respectively. We performed 5 independent replications.

We use a neural network architecture that consists of two hidden layers, with $64$ and $32$ hidden units respectively, for all algorithms. A rectified linear unit (ReLU) is used as the activation function for each hidden layer. The Adam optimizer [36] with a learning rate of $1.0 \times 10^{-4}$ was used. For OTDQN, we use the same parameter settings as in He et al. [25].

For WCDQN, we use a Lagrangian multiplier $\lambda \in [0, 10]$, with a 0.01 discretization. We also used an experience buffer of size 100,000 and initialized it with 10,000 experience tuples that were obtained using a random policy. For the WCDQN algorithm, we set the penalty coefficient $\kappa_U$ to 10, after performing a small amount of manual hyperparameter tuning on the set $\{1, 2, 4, 10\}$.

### D.5 Sensitivity analysis of WCQL with respect to the number of subproblems

We study the performance improvement from WCQL over vanilla Q-learning as the number of subproblems increases for the EV charging problem. We only vary the number of subproblems (from 2 to 5) and keep all other settings as defined in Appendix D.1. The results, given in Table 2, show that the benefits of WCQL become larger as the number of subproblems increases. This provides some additional evidence for the practicality of our approach, especially in regimes where standard methods fail.

Table 2: Cumulative reward and percent improvement of WCQL over QL on the EV-charging problem with a different number of subproblems.

| Algorithm | Number of Subproblems | | | |
|---|---|---|---|---|
| | 2 | 3 | 4 | 5 |
| QL | 5.39 | 6.7 | 5.2 | 3.26 |
| WCQL | 5.35 | 7.14 | 6.28 | 4.66 |
| Percent improvement | -0.7% | 6.6% | 20.8% | 42.9% |

## E Limitations and Future Work

One interesting direction to explore for future work is to address the limitation of learning the Lagrangian upper bound using a fixed and finite set $\Lambda$. Instead, one can imagine the ability to learn the optimal value of $\lambda$ and concentrate the computational effort towards learning the Lagrangian upper bound for this particular $\lambda$, which could potentially lead to tighter bounds. A possible approach is to apply subgradient descent on $\lambda$, similar to what is done in Hawkins [24].

