# OpenReview forum: "Weakly Coupled Deep Q-Networks"
_NeurIPS.cc/2023/Conference — NeurIPS 2023 poster_

### Official Review · Reviewer_QmyG · 2023-07-08

**Soundness:** 3 good
**Presentation:** 4 excellent
**Contribution:** 3 good
**Rating:** 7
**Confidence:** 4

**Summary:**

The paper studies value-based control agents for Weakly Coupled MDPs (WCMDPs), which comprise independent subproblems linked by an action constraint. The authors generalize an existing Lagrangian relaxation approach to action-value methods learned by stochastic approximation to obtain Weakly Coupled Q-Learning (WCQL). Essentially, a Lagrange multiplier is used to create an overestimate of $Q^*$ called $Q^{\lambda*}$; taking the minimum of these two value functions can lead to faster convergence without changing the fixed point. A deep-learning extension of this method relying on a soft constraint added to the DQN loss yields the titular method Weakly Coupled Deep Q-Network (WCDQN). The proposed methods are tested against several well-known baselines (such as DQN and Double DQN) and compare favorably on three WCMDPs: electric vehicle charging (tabular only), multi-product inventory control, and ad matching problems.

**Strengths:**

- Self-contained descriptions of Weakly Coupled MDPs and Lagrangian relaxation, as well as a complete related works discussion, make the paper accessible to people (like me) who do not normally work in this setting. The explanations are clear and well written.
- Interesting theoretical results in the tabular setting. The convergence results look correct and intuitive to me. I checked the proofs in the appendix.
- Interesting, realistic experiment domains and strong empirical results.

**Weaknesses:**

- WCQL relies on discretizing $\lambda \geq 0$ into a finite set of values $\Lambda$ to approximate the upper bound $\min_{\lambda} Q^{\lambda}$. Although this does not affect the ultimate convergence to $Q^*$ in the tabular case, I would imagine that this choice has a significant impact on empirical performance, yet the authors do not discuss this or how to choose the set (that I was able to find at least).
- The computational requirements of the proposed methods are downplayed. The authors state, “We note that, especially in settings where the limiting factor is the ability to collect enough experience, these bounds come nearly for free in that they only require computing a few additional quantities and importantly, they do not require additional experience.” I think this is a strong claim; compared to Q-Learning, at least $N$ additional Q-functions are needed to form the upper bound $Q^{\lambda*}$. In the deep RL setting, only one extra network appears to be needed, in order to generalize over these $N$ functions, but that still doubles the overall computation. Thus, I do not think it is fair to say that the method comes for free; however, the strong empirical results do justify the extra expense.
- Although the tabular theory is sound, the implications of porting it to deep function approximation are unclear and not fully appreciated by the paper in my opinion. The authors say, “the [loss] penalty softly ensures that learned network satisfies the bounds obtained from the Lagrangian relaxation.” However, because the constraint is soft, it does not ensure anything. If the approximated upper bound ever dips below Q*, then the wrong value will be bootstrapped by the network and the learned returns will be suboptimal. Of course, this is inevitable with function approximation (a hard constraint cannot be enforced), but it is worthwhile to clarify in the paper.

**Questions:**

- Why is $\lambda$ called the penalty vector? Is it not a scalar, i.e., the Lagrange multiplier?
- How long (in timesteps) were the agents trained in each of the experiments? Since the figure x-axes use episodes, did agents receive unequal training time?
- How was the penalty coefficient $\kappa_U=10$ chosen?

**Limitations:**

- Need to choose the set $\Lambda$
- Added computational expense
- Upper bound potentially violated with function approximation and soft constraint

---

> ### Author Rebuttal · Authors · 2023-08-10
>
> We thank the reviewer for their time and effort spent providing a thorough review of the paper and for the positive comment on the writing (“Self-contained descriptions of Weakly Coupled MDPs and Lagrangian relaxation, as well as a complete related works discussion, make the paper accessible... The explanations are clear and well written”) and the contributions (“Interesting theoretical results in the tabular setting. The convergence results look correct and intuitive to me. I checked the proofs in the appendix. Interesting, realistic experiment domains and strong empirical results.”).
>
> We address the reviewer’s concerns below. We would highly appreciate it if the reviewer can increase their score if our responses are able to address the reviewer’s concerns.
>
> *WCQL relies on discretizing  into a finite set of values to approximate the upper bound. Although this does not affect the ultimate convergence in the tabular case, I would imagine that this choice has a significant impact on empirical performance, yet the authors do not discuss this or how to choose the set (that I was able to find at least).*
>
> * This is a good point. Under a mild condition, there is way to determine a range [0, \lambda_max], such that the \lambda*, the minimizer of the Lagrangian, falls within this range.
>   * Mild condition: Suppose that for each subproblem i, there is a “zero action” a_i^0 = 0 that contributes zero cost and that all actions contribute non-negative cost (and therefore B(w) >=0 in order for the problem to make sense). Note that in all three of our examples, a zero action exists (the interpretations are “do not charge EV”, “allocate zero resources to the product”, and “do not assign the impression to an advertiser” for the EV charging, multi-product inventory, and online ad-matching problems, respectively), so we believe this is indeed a mild condition for most WCMDPs.
>   * For ease of presentation (not necessary), all subproblem action spaces are {a^0, a^1, …, a^|A|}, where a^k are sorted such that a^k <= a^{k+1}. Also, suppose WLOG that rewards are within [0, R_max].
>   * We claim that we can set \lambda_max = R_max / [(1-\gamma) * a^1]. Here is a sketch of the argument.
>     * Consider the Lagrangian MDP given in equation (3) and consider a particular subproblem i, represented by equation (4).
>     * If we take the “zero-action policy”, i.e., always select a^0 for any state, then the value of this policy is non-negative and independent of \lambda by equation (4): Q_i^{\lambda, \pi_0}(s_i, a^0) >=0.
>     * Next, note that r_i(s_i, a_i) + \gamma E[max Q_i^\lambda(s’_i, a’_i)] <= R_max / (1-\gamma) for any \lambda, s_i, and a_i due to the nonnegativity of both \lambda and a. Therefore, for any action a^k where k >= 1 (i.e., all actions except the zero action), we have that Q_i^\lambda(s_i, a^k) < 0 whenever \lambda > \lambda_max.
>     * However, we know the “zero-action policy” achieves non-negative value for each subproblem and that this value is independent of lambda. Hence, looking at equation (5), we see that Q^\lambda(s, a) must be increasing in \lambda for all \lambda > \lambda_max. This means that we only need to consider [0, \lambda_max] to find the minimizer of the Lagrangian.
>
> * We would be happy to include a discussion and this result in the paper / appendix.
>
> *The computational requirements of the proposed methods are downplayed. The authors state, “... these bounds come nearly for free in that they only require computing a few additional quantities and importantly, they do not require additional experience.” I think this is a strong claim; compared to Q-Learning, at least additional Q-functions are needed to form the upper bound. In the deep RL setting, only one extra network appears to be needed, in order to generalize over these functions, but that still doubles the overall computation. Thus, I do not think it is fair to say that the method comes for free; however, the strong empirical results do justify the extra expense.*
>
> * Thanks for your comment. We agree with you -- our intention was to primarily communicate that our method does not require additional experience, but we now see that this was unclear. We will fix the wording as follows: “We note that, especially in settings where the limiting factor is the ability to collect enough experience, one can trade-off extra computation to derive these bounds and improve RL performance without the need to collect additional experience.”
>
> *Although the tabular theory is sound, the implications of porting it to deep function approximation are unclear and not fully appreciated by the paper in my opinion... The authors say, “the [loss] penalty softly ensures that learned network satisfies the bounds obtained from the Lagrangian relaxation.” However, because the constraint is soft, it does not ensure anything...worthwhile to clarify in the paper.*
>
> * Noted and we agree. Our intent was to acknowledge this using the wording "softly ensures", but now realize that this wording is confusing. We will edit this as you suggested.
>
> *Why is $\lambda$ called the penalty vector? Is it not a scalar, i.e., the Lagrange multiplier?*
>
> * Yes, you are correct $\lambda$ is a scalar, not a vector. We will correct this typo in the paper.
>
> *How long (in timesteps) were the agents trained in each of the experiments? Since the figure x-axes use episodes, did agents receive unequal training time?*
>
> * Please note that all agents were trained for the same number of time-steps (EV charging 300,000; multi-product inventory control problem 125,000; online-stochastic ad matching 300,000 steps)
>
> *How was the penalty coefficient $\kappa_U$ chosen?*
>
> * We performed a small amount of manual hyperparameter tuning, trying a few coefficients {1,2,4,10}, and 10 seemed to have the best performance. In general, because this penalty coefficient is over a one-dimensional space, our recommendation is to conduct a simple grid search. We will add this recommendation to the paper.

---

> > ### Comment · Reviewer_QmyG · 2023-08-16
> >
> > Thank you for the detailed clarifications! Adding these points of discussion to the paper will help others understand your algorithm design and experiment choices better. I will keep my score the same and continue to recommend this paper for acceptance.

---

> > > ### Author Response · Authors · 2023-08-16
> > >
> > > We appreciate the response and your time spent reviewing the paper! Please let us know if any additional questions arise.

---

### Official Review · Reviewer_da8o · 2023-07-10

**Soundness:** 3 good
**Presentation:** 3 good
**Contribution:** 3 good
**Rating:** 6
**Confidence:** 3

**Summary:**

The authors present a new algorithm, "WCDQN" for solving structured problems know as weakly coupled MDPs. The method dynamically estimates upper bounds via Lagrangian relaxation essentially to improve the value of collected experience across the structured subproblems. The authors present a (re)formulation of the problem via the dual bound of the linking constraints and numerically demonstrate the effectiveness of the method on a set of WCMDP problems.

**Strengths:**

1. Nicely written, enjoyable reading
2. Relatively simple and clear problem formulation


**Weaknesses:**

1. moderate ablation. the proposed method allows for a decomposition based on the number of "subproblems" but the relative effect to this key variable is not explicitly tested

**Questions:**

1. It would seem to be the case that this method should scale significantly better for problems with specific substructure. Why not demonstrate this explicitly with a custom problem / domain to validate the extreme case?
2. [note] I think Fig3 is missing a y-axis?
3. [note] your compilation resulted in non-clickable links,

**Limitations:**

NA - nothing in addition to what is present in the actual mathematics.

---

> ### Author Rebuttal · Authors · 2023-08-10
>
> We thank the reviewer for their kind comments (“Nicely written, enjoyable reading", "relatively simple and clear problem formulation”). It seems that the main concern that the reviewer had is due to the lack of an experiment showing the effect of the number of subproblems on the solution quality. We have now performed this experiment (details below) and hope that we addressed the reviewer’s concern. If so, we’d appreciate it if the reviewer could consider raising their score.
>
> *Moderate ablation. the proposed method allows for a decomposition based on the number of "subproblems" but the relative effect to this key variable is not explicitly tested*
>
> * Regarding the reviewer’s concern about the relative effect of the algorithm with respect to the number of subproblems, this is a great point. We’ve just completed a new experiment that studies the performance improvement from WCQL over vanilla QL as the number of subproblems increases. The **results are in the attached PDF of the global response** and validate our hypothesis that the benefits of WCQL become larger as the number of subproblems increases. This provides some additional evidence for the practicality of this approach, especially in regimes where standard methods fail.
>
> *It would seem to be the case that this method should scale significantly better for problems with specific substructure. Why not demonstrate this explicitly with a custom problem / domain to validate the extreme case?*
>
> * We have created a new figure to visualize the type of weakly-coupled structure for which our method is designed; **please see the attached PDF of the global response**. The visualization shows that a primary agent needs to make decisions for $N$ **nearly independent subproblems** (represented by red, blue, and yellow), except that the decisions must satisfy a global resource constraint. In our view, all of the numerical examples we provided (EV charging, multi-product inventory, and online ad-matching), are already specifically selected because they enjoy such structure:
>   * In the EV charging problem, each charging spot is a subproblem, but the EV charging manager needs to make sure the overall electricity cost at any given time does not exceed a certain amount.
>   * In the multi-product inventory problem, each type of product being manufactured is its own subproblem, but the decision maker needs to make sure that the manufacturing resources used at each time period does not exceed a particular budget.
>   * In the online ad-matching problem, each advertiser is a subproblem, but the ad company can only assign each impression to only one advertiser.
> * Therefore, we are not sure what kind of custom problem we can add. Did the reviewer have something specific in mind? Please let us know if we misunderstood the question.
>
> *I think Fig3 is missing a y-axis?*
>
> * In Fig 3, the y-axis are the Q-values. Sorry that this was mistakenly removed. We will add the y-label to the figure to make it clear.
>
> *Your compilation resulted in non-clickable links.*
>
> * Noted, thanks! We will fix this in the final paper.

---

### Official Review · Reviewer_HJ3Y · 2023-07-10

**Soundness:** 3 good
**Presentation:** 4 excellent
**Contribution:** 3 good
**Rating:** 5
**Confidence:** 3

**Summary:**

This paper considers the problem of learning optimal policies for a weakly coupled Markov decision process (WCMDP). The paper extends the classical DQN algorithm to this setting by introducing separate lagrangian relaxations. Experiments are conducted on 3 operational research testbeds.

**Strengths:**

This paper has a very complete structure, with a detailed problem introduction, an insightful algorithm derivation, a theoretical analysis as well as solid numerical studies. The presentation is particularly clear and easy to follow. The algorithm is intuitive and convincing. You cannot really find any flow from this paper.

**Weaknesses:**

My biggest concern about this work is the audience. As a practitioner in deep RL, this is literally the first time for me to hear the problem of WCMDP. Although I do appreciate that this work teaches me a lot, I don't know whether this paper can attract the attention of the deep RL community, particularly considering the test domains are pretty toy. This paper looks to me more like a work that brings deep RL techniques to the OR community rather than the reverse direction.

**Questions:**

N/A

**Limitations:**

The significance of the problem for the deep learning community.

---

> ### Author Rebuttal · Authors · 2023-08-10
>
> We thank the reviewer for acknowledging the strengths of our paper: “This paper has a very complete structure, with a detailed problem introduction, an insightful algorithm derivation, a theoretical analysis as well as solid numerical studies. The presentation is particularly clear and easy to follow. The algorithm is intuitive and convincing. You cannot really find any flaw from this paper.” We appreciate these positive comments and they mean a lot to us. Thank you again.
>
> We understand that the main reason for our score of 5 is due to a somewhat philosophical question of whether WCMDPs fit into the NeurIPS conference. We provide our rebuttal to this point below and hope that the reviewer is willing to consider raising the score, especially in light of the reviewer’s strong positive comments regarding the technical strengths of the paper.
>
> Below is our response.
>
> *My biggest concern about this work is the audience. As a practitioner in deep RL, this is literally the first time for me to hear the problem of WCMDP. Although I do appreciate that this work teaches me a lot, I don't know whether this paper can attract the attention of the deep RL community, particularly considering the test domains are pretty toy. This paper looks to me more like a work that brings deep RL techniques to the OR community rather than the reverse direction.*
>
> * We would like to make two main points for why we believe WCMDPs would be valuable to the deep RL, and more generally, the NeurIPS community.
>
>   1. First, the NeurIPS community has recently shown a strong interest in moving beyond the traditional domains (Atari, board games, MuJoCo, etc) into arguably more practical settings. For example, the RL4RealLife Workshop at NeurIPS 2022 featured talks on inventory management, manufacturing, finance, energy demand response, traffic signal control, data center optimization, commercial cooling systems, power grid optimization, healthcare, and more. Most of these topics were originally studied in the OR community. Leading AI labs like Deepmind have also shown an interest in data center cooling optimization (another OR problem).
>   2. Second, weakly coupled MDPs are highly relevant to problems in online advertising, which is a domain that is of interest to the NeurIPS community. The paper Boutilier and Lu (2006), published in UAI and authored by Google researchers, makes use of a weakly coupled MDP for the problem of budget optimization in online advertising. In addition, many papers on restless bandits (which are very closely related to weakly coupled MDPs) have been published in AI conferences; two such examples are Mate et al., 2020 (NeurIPS) and Killian et al., 2022 (UAI), suggesting that there is existing interest in these models from AI researchers.
>
> * We hope that the reviewer agrees that the problems we study in this paper are real-world problems with huge state (continuous and up to 10 dimensions) and action spaces (3^10), each with significant practical interest. We believe this work will have a positive impact on the deep RL community and hopefully leads to or encourages a shift toward solving practical, real-world problems.

---

> > ### Comment · Reviewer_HJ3Y · 2023-08-12
> >
> > Thanks for the clarifications. I will keep a positive perspective on this work and leave the final decision to the discussion phase.

---

> > > ### Author Response · Authors · 2023-08-13
> > >
> > > Thank you for reading our rebuttal and for the kind comments!
> > >
> > > Please let us know if you have any further questions about the applicability of the weakly-coupled technique to real-world problems that the deep RL community is interested in. We do believe that we are both bringing an OR technique (the weakly-coupled formulation + the decomposition and upper bound) to the deep RL community, while also bringing deep RL ideas (DQN agents) to the OR community and hope that this paper can be beneficial to both sides.

---

### Official Review · Reviewer_aL5V · 2023-07-12

**Soundness:** 3 good
**Presentation:** 2 fair
**Contribution:** 3 good
**Rating:** 5
**Confidence:** 3

**Summary:**

This paper introduces novel Q-learning based approaches (WCQL and WCDQN) to solving weakly-coupled Markov decision processes. The approaches work by running Q-learning for the sub-problems of the WCMDP to construct an upper bound on the optimal Q-values of the overall problem. The WCMDP constraint is incorporated into the reward structure of the task via approximate Lagrangian relaxation. In the function approximation case, the constraint is enforced as a penalty to the loss. A theoretical analysis is performed for the tabular case and experiments are run to demonstrate the performance of both the tabular and DQN-based approaches.

**Strengths:**

- The paper is generally easy to follow and clearly written, and the problem setting is well-motivated.

- The proposed approaches appear to be simple and relatively original.

- If this is indeed the first model-free approach to WCMDPs, it represents a significant advance for this problem setting.

**Weaknesses:**

- The core weakness of both the theoretical and empirical analyses, in my opinion, is the lack of treatment of constraint violations by WCQL and WCDQN. The theoretical analyses, as far as I can tell, assume a fixed, arbitrary Lagrange multiplier and show convergence of the Q-values to the optimal Q-values for *that choice of Lagrange multiplier*. This, however, says nothing about the resulting policy’s constraint violations. Similarly, in the experiments, there are no results showing the constraint violations of each method. Indeed, the “soft constraint” formulation seems strange for problems like the EV task, for in which in practice assigning too many cars to too few charging ports would seem to be a serious problem. I think in particular it would be helpful to add plots showing constraint violations for each method and whether they find feasible policies.

- Related to the above, there’s no discussion of whether even in an idealized case where $\lambda$ was a continuous variable the proposed approach converges to a saddle point of the Lagrangian (i.e., actually solves the “true” optimization problem), even if in practice discretization is not a significant detriment to performance.

- As far as I can see, there is no discussion of the limitations of the proposed approaches.

- The experimental evaluation is fairly limited.

- More details on the baselines in the main text would be helpful to improve clarity. I would suggest changing the citations to a numerical format to save space.

Minor:
- The stochastic gradient update at the end of Section 3.2 would ideally reflect the update of minibatch SGD rather than pure SGD.

- The $\wedge$ symbol in Eq. 11 seems unnecessary, it would be more clear to just write $\min(\cdot, \cdot)$.

- From a clarity perspective, it would be helpful if the initial description of a WCMDP contained an example to anchor readers’ understanding.

- Figure 3A has no y-axis label. Also, I think it would be more informative to plot the upper bound on average across $(s,a)$ pairs rather than at only one point.

- It would make sense to add distributional RL [1] and Rainbow [2] to the list of extensions of DQN in the introduction or related work sections. The authors should also contrast their setting to constrained MDPs [3].


I think this is interesting work, and I am open to increasing my score based on the degree to which the above concerns are addressed.


[1] https://arxiv.org/abs/1707.06887

[2] https://arxiv.org/abs/1710.02298

[3] https://www-sop.inria.fr/members/Eitan.Altman/TEMP/h.pdf

**Questions:**

- Does the sub-problem action-value network parameterization mean that all of the subproblem action spaces have to have the same number of actions?

- Is there some intuition as to why double Q-learning performs worse than regular Q-learning? I would think the fact that double Q-learning attempts to reduce value overestimation would have a similar effect to ensuring the overall task Q-value estimate doesn’t exceed the estimated upper bound.

**Limitations:**

The authors do not appear to discuss any limitations of their method (see above). I do not directly see any negative societal implications of this work, though in practice it would depend on the problem to which it is applied and any guarantees (or lack thereof) regarding constraint violations.

---

> ### Author Rebuttal · Authors · 2023-08-10
>
> We thank the reviewer for recognizing the originality (“proposed approaches appear to be simple and relatively original”) and the importance of our work (“the problem setting is well-motivated”, “represents a significant advance for this problem setting”). We’d appreciate it if the reviewer would consider increasing their score if the comments are satisfactory. All minor comments are noted and will be fixed.
>
> *The core weakness... in my opinion, is the lack of treatment of constraint violations ...*
>
> * Constraint violation: we believe there is a little misunderstanding here. We would like to clarify that there are two types of “constraints” used in this paper (unrelated to each other):
>   1. Action-selection constraints that are central to the original problem
>   2. Q-value constraints used by WCDQN to speed up the learning process
> * These two types of constraints are two separate notions in our paper. We assume that by “policy’s constraint violations”, the reviewer is referring to the action-selection constraints (type 1). However, note that the action selection is always feasible: please see assumption 1 (observability of b(w) once the agent lands in a state, which is enough information for the agent to compute the set of feasible actions). Therefore, the action-selection constraints are never violated. In lines 183-186 in the paper, we give further context on Assumption 1: “the agent learning on the main problem will always sample a feasible action. In each of our example applications, Assumption 1 naturally holds: e.g., in the EV charging problem, there are a limited number of available charging stations b(w).”
> * Regarding the comment “soft constraint seems strange for the EV task”, we believe this is a misunderstanding, since the use of “soft constraints” is within the WCDQN algorithm and is not related to the action-selection constraints. The main idea behind WCDQN is to learn the Lagrangian Q-value (which is an upper bound on Q*) and use this upper bound to encourage our Q-estimate to be close to Q* via soft constraints.
>
> *there’s no discussion of whether even in an idealized case where \Lambda...*
> * Thanks for your question. We have two interpretations of your comment, depending on what you mean by “true” optimization problem. If by “true optimization problem,” you are referring to the problem of minimizing over \lambda, then yes, in the ideal situation, we are able to solve it to optimality. This is because Q^\lambda is a piecewise linear and convex function of \lambda (Adelman and Mersereau, 2008) and we can in principle find its minimizer using standard convex optimization techniques, such as subgradient descent.
> * If by “true optimization problem,” you are referring to the original MDP, then we believe you are asking about the duality gap between the Lagrangian formulation vs the optimal value of the MDP. One highly relevant paper is Adelman and Mersereau (2008), which provides two results in this direction (Theorems 2 and 4). The former gives (somewhat stringent) conditions for when the Lagrangian relaxation is tight and the latter gives a bound on the duality gap for more general conditions.
>   * We will add discussions of both of these results into the paper’s appendix. Note that our approach makes use of the fact that Q^\lambda is an upper bound of Q* to speed up the learning of Q* (by encouraging the estimate of Q to be below the estimate of the upper bound), but our approach does not critically depend on the upper bound being tight. As long as the bound is somewhat good, we expect to see empirical benefits.
>
> *there is no discussion of the limitations..*
>
> * The main limitation in our opinion is that the finite set of \lambda values needs to be specified as input to the algorithm. In future work, we would like to explore the idea of adaptively adjusting \lambda (perhaps using a finite-difference subgradient), which would remove the need of estimating Q^\lambda for many \lambdas.
>
> *experimental evaluation is fairly limited.*
>
> * We’ve completed a new experiment to study the effect of the number of subproblems; please see the **global response.**
>
> *More details on the baselines in the main text...*
>
> * Noted and will update!
>
> *subproblem action spaces have to have the same number of actions?*
>
> * To address different subproblem action spaces, we can change the network architecture to receive the state-action pair (s_i, a_i) as input and output the predicted Q-value. This simple change does not interfere or affect WCDQN's main idea. Will address in paper.
>
> *intuition as to why double Q-learning performs worse than regular Q-learning?*
>
> * Great question! Note that both double Q-learning and Q-learning are both operating directly on the joint state space, making for a difficult exploration problem. Our belief is that due to Q-learning’s larger overestimation bias, this may sometimes lead to better exploration properties (if those overestimated states happen to be of high value). See the Maximin Q-learning paper of Lan et al., 2020 for discussion of this phenomenon (https://openreview.net/pdf?id=Bkg0u3Etwr).
>
> * Intuition on how WCQL and double QL handle overestimation and why WCQL works better:
>   * For double Q-learning, the overestimation is handled on a state-by-state basis.
>   * For WCQL, suppose we have 3 subproblems, each subproblem with state space (1, 2, 3), leading to 3^3 total states. Now, even if we just visit 3 of these states (1, 1 ,1), (2, 2, 2), and (3, 3, 3), then due to the properties of the decomposition, we have information for all subproblem states of all subproblems. Hence, despite only observing 3 of 27 states, we are able to apply the upper-bound to any arbitrary state in the state space. The intuition here is that WCQL enables a sort of generalization to unseen states by leveraging weakly-coupled structure. This is the main reason for WCQL’s effectiveness. We hope that this example and discussion helps to address your question.

---

> > ### Comment · Reviewer_aL5V · 2023-08-12
> > **Response**
> >
> > Thank you to the authors for your detailed rebuttal! It has clarified a number of my questions. I think the added discussion should will benefit the text as well. I'm happy to raise my score.

---

> > > ### Author Response · Authors · 2023-08-13
> > >
> > > Thank you for the acknowledging that we have helped to clarify some questions and for raising the score! We appreciate your time. Please let us know if we can further expand or clarify any points that we were not able to fully discuss in the rebuttal. We'd love to have the opportunity to address any remaining concerns you have in the next few days before the discussion period ends.

---

### Author Rebuttal · Authors · 2023-08-10

We thank all reviewers for their thoughtful comments and suggestions. Since receiving the reviews, we have completed a new experiment that studies the performance improvement from WCQL over vanilla QL as the number of subproblems increases, as suggested by Reviewer da8o. This is a great suggestion and validates our hypothesis that the benefits of WCQL become larger as the number of subproblems increases. The new results provide further evidence regarding the practicality of WCQL/WCDQN, especially in regimes where standard methods fail. The results are shown in the attached PDF.

We also included a visualization of a weakly-coupled problem in the attached PDF as a response to a question by Reviewer da8o. However, we hope that the visualization can also be useful to all readers to give them a better understanding of weakly-coupled problem structure.

---

### Decision · Program_Chairs · 2023-09-21

**Decision:**

Accept (poster)

**Comment:**

After discussion with the authors, all the reviewers are recommending to accept the paper. This is also my recommendation.